# LINEAR COMBINATIONS OF LATENTS IN GENERATIVE MODELS: SUBSPACES AND BEYOND

Erik Bodin[1]  Alexandru Stere[4]  Dragos D. Margineantu[5]

Carl Henrik Ek[1,3]  Henry Moss[1,2]

[1]University of Cambridge  [2]Lancaster University  [3]Karolinska Institutet

[4]Boeing Commercial Airplanes  [5]Boeing AI

## ABSTRACT

Sampling from generative models has become a crucial tool for applications like data synthesis and augmentation. Diffusion, Flow Matching and Continuous Normalising Flows have shown effectiveness across various modalities, and rely on latent variables for generation. For experimental design or creative applications that require more control over the generation process, it has become common to manipulate the latent variable directly. However, existing approaches for performing such manipulations (e.g. interpolation or forming low-dimensional representations) only work well in special cases or are network or data-modality specific. We propose Latent Optimal Linear combinations (LOL) as a general-purpose method to form linear combinations of latent variables that adhere to the assumptions of the generative model. As LOL is easy to implement and naturally addresses the broader task of forming **any** linear combinations, e.g. the construction of subspaces of the latent space, LOL dramatically simplifies the creation of expressive low-dimensional representations of high-dimensional objects.

## 1 INTRODUCTION

Generative models are a cornerstone of machine learning, with diverse applications including image synthesis, data augmentation, and creative content generation. Diffusion models (Ho et al., 2020; Song et al., 2020a;b) have emerged as a particularly effective approach to generative modeling for various modalities, such as for images (Ho et al., 2020), audio (Kong et al., 2020), video (Ho et al., 2022), and 3D models (Luo & Hu, 2021). A yet more recent approach to generative modelling is Flow Matching (Lipman et al., 2022), built upon Continuous Normalising Flows (Chen et al., 2018), that generalises diffusion to allow for different probability paths between data and latent distribution, e.g. defined through optimal transport (Gulrajani et al., 2017; Villani et al., 2009).

As well as generation, these models allow inversion where, by running the generative procedure in the opposite direction, data objects can be transformed deterministically into a corresponding realisation in the latent space. Such invertible connections between latent and data space provide a convenient mechanism for controlling generated objects by manipulating their latent vectors. The most common manipulation is to attempt semantically meaningful interpolation of two generated objects (Song et al., 2020a;b; Luo & Hu, 2021) by interpolating their corresponding latent vectors. However, the optimal choice of interpolant remains an open question, with simple approaches like linear interpolation leading to intermediates for which the model fails to generate plausible objects.

White (2016) argues that poor quality generation under linear interpolation is due to a mismatch between the norms of the intermediate vectors and those of the Gaussian vectors that the model has been trained to expect. Indeed, it is well-known that the squared norm of a $D$-dimensional unit Gaussian follows the chi-squared distribution. Consequently, likely samples are concentrated in a tight annulus around a radius $\sqrt{D}$ — a set which is not closed under linear interpolation. Because of this it is common to rely instead on spherical interpolation (Shoemake, 1985) (SLERP) that maintains similar

norms as the endpoints. Motivated by poor performance of interpolation between the latent vectors provided by inverting natural images, alternatives to SLERP have recently been proposed (Samuel et al., 2023; Zheng et al., 2024). However, these procedures are costly, have parameters to tune, and — like SLERP — are challenging to generalise to other types of manipulations beyond interpolation, such as constructing the subspaces needed for latent space optimisation methods (Gómez-Bombarelli et al., 2018), or adapting to more diverse latent distributions beyond Gaussian.

In this work, we demonstrate that successful generation from latent vectors is strongly dependent on whether their broader statistical characteristics match those of samples — a stronger condition than just having e.g. likely norms. We show that effective manipulation of latent spaces can be achieved by following the simple guiding principle of *adhering to the modelling assumptions of the generation process*. Our primary contributions are as follows:

- We show that even if a latent leads to a valid object upon generation, it can fail to provide effective interpolation if it lacks the typical characteristics of samples from the latent distribution — as diagnosed by statistical distribution tests.
- We introduce Latent Optimal Linear combinations (LOL) — an easy-to-implement method for ensuring that interpolation intermediates continue to match the latent distribution.
- We show that LOL — a closed-form transform that makes no assumptions about the network structure or data modality — constitutes a Monge optimal transport map for linear combinations to a general class of latent distributions.
- We demonstrate that LOL can be applied to define general linear combinations beyond interpolations, including centroids (Figure 2) and, in particular, to define meaningful low-dimensional latent representations (Figure 1) — a goal achieved previously with significant expense and only for specific architectures.

  Implementation and examples: https://github.com/bodin-e/linear-combinations-of-latents

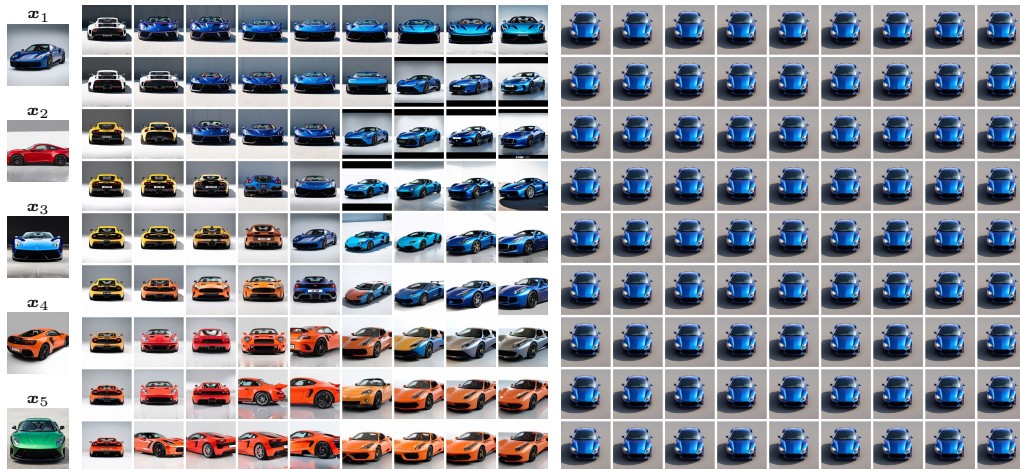

Figure 1: **Low-dimensional latent subspaces**. A 5-dimensional subspace from the flow matching model Stable Diffusion 3 (Esser et al., 2024) extracted using LOL (left) from the latents corresponding to images $x_1, \ldots, x_5$. The left plot show generations from uniform grid points across an axis-aligned slice of the subspace coordinate system, centered around the coordinate for $x_1$. Each coordinate in the subspace correspond to a linear combination of latents, which define basis vectors. The right plot shows the corresponding subspace *without* the proposed LOL transformation. See Figure 6, Figure 7 and Section G in the appendix for additional examples.

## 2 BACKGROUND

### 2.1 GENERATIVE MODELLING WITH LATENT VARIABLES

The methodology in this paper applies to any generative model that transforms samples from a known distribution in order to model another distribution — typically, a complicated distribution of high-

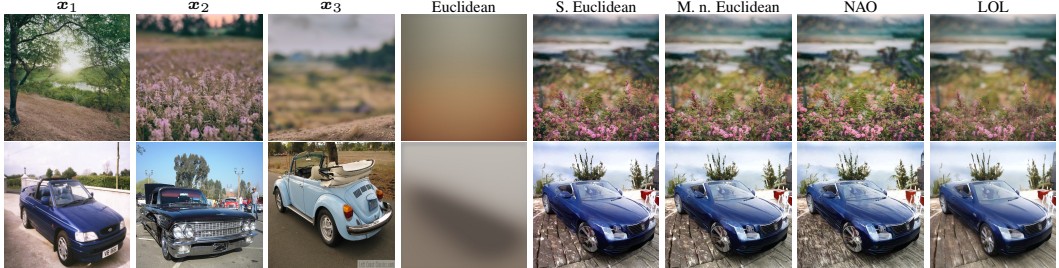

Figure 2: **Centroid determination**. Generation using Stable Diffusion 2.1 (Rombach et al., 2022) from the centroid of the latents corresponding to images $\boldsymbol{x}_1$, $\boldsymbol{x}_2$, $\boldsymbol{x}_3$ using different methods. Note that our proposed method removes several artifacts, such as unrealistic headlights and chassi texture.

dimensional objects. For clarity of exposition, we focus on popular Diffusion and Flow Matching models using Gaussian latents, and then extend to general latent distributions in the appendix.

**Diffusion models** learn a process that reverses the effect of gradually adding noise to training data (Ho et al., 2020). The noise level is governed by a schedule designed so that the noised samples at the final time index $T$ follow the latent distribution $\boldsymbol{x}^{(T)} \sim p(\boldsymbol{x})$. To sample from the diffusion model, one starts by drawing a sample from the latent distribution before iteratively evaluating the reverse process with learnt parameters $\boldsymbol{\theta}$, denoising step-by-step, generating a sequence $\{\boldsymbol{x}^{(T)}, \boldsymbol{x}^{(T-1)}, \ldots, \boldsymbol{x}^{(0)}\}$

$$\boldsymbol{x}^{(t-1)} \sim p_{\boldsymbol{\theta}}(\boldsymbol{x}^{(t-1)} \mid \boldsymbol{x}^{(t)}),$$

finally producing a generated object $\boldsymbol{x}^{(0)}$. Diffusion has also been extended to the continuous time setting by Song et al. (2020b), where the diffusion is expressed as a stochastic differential equation. Note that, by using the Denoising Diffusion Implicit Model (DDIM) (Song et al., 2020a) or the probability flow formulation in Song et al. (2020b), the generation process can be made deterministic, i.e. the latent representation $\boldsymbol{x}^{(T)} \sim p(\boldsymbol{x})$ completely specifies the generated object $\boldsymbol{x}^{(0)}$.

**Flow Matching** Lipman et al. (2022) is an efficient approach to train Continuous Normalising Flows (CNF) Chen et al. (2018) — an alternative class of generative models that builds complicated distributions from simple latent distributions using differential equations. CNFs model the evolution of data points over continuous time using an ordinary differential equation (ODE) parameterized by a neural network, providing a deterministic relationship between latent and object space. Mathematically, the transformation of a latent sample $\boldsymbol{x}^{(T)} \sim p(\boldsymbol{x})$ at time $t = T$ to $\boldsymbol{x}^{(t)}$ at time $t$ is governed by $\boldsymbol{f}(\boldsymbol{x}^{(t)}, t; \boldsymbol{\theta})$, where $\boldsymbol{f}(\cdot; \boldsymbol{\theta})$ is a neural network with parameters $\boldsymbol{\theta}$. Flow matching allows the transformation dynamics of the CNF to be learnt without needing to simulate the entire forward process during training, which improves stability and scalability.

Note that both diffusion and flow matching models can generate in a deterministic manner, where the realisation of the latent variable completely specifies the generated data object, e.g. the image. Moreover, by running their deterministic generation formulation in reverse, one can obtain the corresponding latent representation associated with a known object, referred to as "inversion". In this paper we focus on the effect of properties of the latent vectors, and their manipulation, on the generation process. For simplicity of notation, henceforth we drop the time index and refer to the latent variable $\boldsymbol{x}^{(T)}$ as $\boldsymbol{x}$, and use subscript indexing to denote realisations of the latent variable.

## 2.2 INTERPOLATION OF LATENTS

**Linear Interpolation**. The most common latent space manipulation is interpolation, where we are given two latent vectors $\boldsymbol{x}_1$ and $\boldsymbol{x}_2$, referred to as *seed latents* (or seeds), and obtain intermediate latent vectors. The simplest approach is to interpolate linearly between the seeds to get intermediates

$$\boldsymbol{y}_{\text{lin}}^w = w\boldsymbol{x}_1 + (1 - w)\boldsymbol{x}_2 \quad \text{for} \quad w \in [0, 1].$$

**Spherical Interpolation**. However, as discussed in (White, 2016) in the context of Variational Autoencoders (Kingma, 2013) and Generative Adversarial Networks (Goodfellow et al., 2014), linear interpolation yields intermediates with unlikely norms for Gaussian samples and results in highly implausible generated objects (e.g. all-green images). Consequently, it is common to instead use

spherical interpolation (SLERP) (Shoemake, 1985), which instead builds interpolants via

$$\boldsymbol{y}_{\text{SLERP}}^w = \frac{\sin w\theta}{\sin \theta} \boldsymbol{x}_1 + \frac{\sin((1-w)\theta)}{\sin \theta} \boldsymbol{x}_2 \quad \text{for} \quad w \in [0,1] \quad \cos \theta = \frac{\langle \boldsymbol{x}_1, \boldsymbol{x}_2 \rangle}{||\boldsymbol{x}_1||\,||\boldsymbol{x}_2||},$$

to maintain similar norms for the intermediates as the endpoints. SLERP has become popular and is the standard method for interpolation also in the more recent models (Song et al., 2020a;b).

**Norm-aware Optimisation**. Motivated by poor performance of SLERP when interpolating between the latent vectors corresponding to inverted natural images (rather than using those sampled directly from the model's latent distribution), Zheng et al. (2024) propose to add additional Gaussian noise to interpolants via the inversion procedure to control a trade-off between generation quality and adherence to the seed images (i.e. the images corresponding to the latents at the interpolation endpoints). Alternatively, Samuel et al. (2023) advocate for Norm-Aware Optimisation (NAO) which uses back-propagation to identify interpolation paths $\gamma : [0,1] \to \mathbb{R}^d$ that solve

$$\inf_{\gamma} - \int \log \mathbb{P}(\gamma(s)) ds \quad \text{s.t.} \quad \gamma(0) = \boldsymbol{x}_1, \gamma(1) = \boldsymbol{x}_2,$$

where $\log \mathbb{P} : \mathbb{R}^d \to \mathbb{R}$ is the log likelihood of the squared norm under its sampling distribution $\chi^2(D)$ for unit Gaussian samples. NAO can also calculate centroids of $K$ latents by simultaneous optimisation of paths between the centroid and each respective latent.

All the proposed interpolation methods above aim to make the intermediates adhere to statistics of Gaussian samples, however, require significant computation, have hyperparameters to tune, or lack theoretical guarantees. Moreover — in common with spherical interpolation — they are difficult to generalise beyond Gaussian latent variables or to more general expressions of the latents beyond interpolation, such as building subspaces based on their span — as required to build expressive low-dimensional representations.

In this work we will propose a simple technique that guarantees that interpolations follow the latent distribution if the original (seed) latents do, applies to a general class of latent distributions, and enables us go beyond interpolation. Moreover, we will address how to assess the distributional assumption of the seed latents, to simplify the diagnosis of errors at the source.

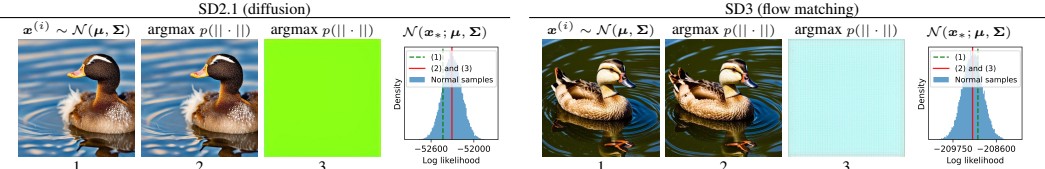

Figure 3: **Likelihood and norm insufficient**. Column (1) of each panel shows an image generated using a random sample from the associated Gaussian latent distribution for the diffusion model of Rombach et al. (2022) (left side) and the flow matching model of Esser et al. (2024) (right side). Columns (2) and (3) both show images generated from latents with the most likely norm according to their respective latent distribution. Columns (2) use the same Gaussian samples as in columns (1) but rescaled to have this norm, also yielding realistic images. Meanwhile, columns (3) show the failed generation from constant vectors $s\boldsymbol{I}$ scaled to have the most likely norm according to the latent distribution but lacking other characteristics (e.g. not having all-equal values) that the network was trained to expect, even though its likelihood $\mathcal{N}(s\boldsymbol{I}; \boldsymbol{\mu}, \boldsymbol{\Sigma})$ is typical of real samples. Moreover, the distribution mode $\boldsymbol{\mu}$, which also lacks needed characteristics , has vastly higher log likelihood than any realistic sample; -33875 and -135503 for the two models, respectively. See Table 2 for an example of failed generation using the mode $\boldsymbol{\mu}$.

## 3 ASSESSING VALIDITY OF LATENTS VIA DISTRIBUTION TESTING

Before introducing our proposed method, we first explore evidence for our central hypothesis that tackles a misconception underpinning current interpolation methods for Gaussian latents:

*Having a latent vector with a norm that is likely for a sample from the latent distribution $\mathcal{N}(\boldsymbol{\mu}, \boldsymbol{\Sigma})$ is not a sufficient condition for plausible sample generation. Rather, plausible generation requires*

*a latent vector with characteristics that match those of a sample from $\mathcal{N}(\boldsymbol{\mu}, \boldsymbol{\Sigma})$ more generally (as evaluated by normality tests), with a likely norm being only one such characteristic.*

The characteristics of the random variable $\boldsymbol{x}$ may be described by a collection of statistics, e.g. its mean, variance or norm. Above, we hypothesise that if a specified latent $\boldsymbol{x}_*$ has an extremely unlikely value for a characteristic for which the network has come to rely, then implausible generation will follow. A norm is unlikely if it has a low likelihood under its sampling distribution; for unit Gaussians it is $\chi(D)$, see Section 1. While the norm has been shown to often be an important statistic (Samuel et al., 2023), our Figure 3 illustrates on a popular diffusion model (Rombach et al., 2022) and a flow matching model (Esser et al., 2024) that a latent with a likely norm — even the most likely — can still induce failure in the generative process if other evidently critical characteristics are unmet.

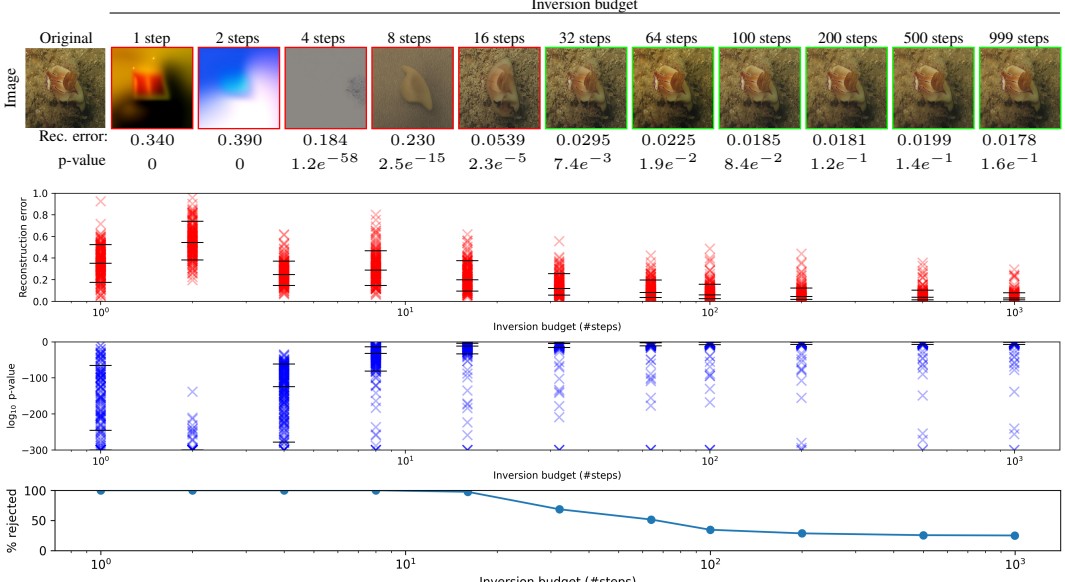

Figure 4: **Normality testing of latent vectors obtained from inversion**. We show the LPIPS (Zhang et al., 2018) reconstruction errors (second row, in red) of 200 inverted randomly selected images across 50 random classes from ImageNet1k (Deng et al., 2009), the p-values of their inversions (third row), and rejection rates (bottom row) of the Kolmogorov-Smirnov normality test applied to the corresponding latent obtained from inversion under various step budgets. We use the diffusion model of Rombach et al. (2022), always using its maximum number of steps (999) for generation, and denote the 10th, 50th and 90th percentiles with black lines. The first row shows image reconstructions using its inversion at each budget, highlighted in red when the latent was rejected ($p < 1e^{-3}$), with the interpretation that the characteristics of the latent were unlikely for a real sample according to the KS test. We note the strong correlation between inversion budgets providing low reconstruction errors and those for which the p-values of the latents are realistic — taking values likely to occur by chance for real samples. However, as we will see in Figure 5, there are still many latents with low reconstruction error yet extremely low p-value, and this often severely affects the quality of its interpolants.

In general, neural networks have many degrees of freedom, and so it is difficult to determine and list the input characteristics that a specific model relies on. Therefore, rather than trying to determine all necessary statistics for each model, we propose instead to rely on standard methods for distribution testing, a classic area of statistics (Razali et al., 2011; Kolmogorov, 1933; Shapiro & Wilk, 1965) — here exploring the (null) hypothesis that a latent vector $\boldsymbol{x}_* \in \mathbb{R}^D$ is drawn from $\mathcal{N}(\boldsymbol{\mu}, \boldsymbol{\Sigma})$. Popular normality tests consider broad statistics associated with normality. For example, the Kolmogorov–Smirnov (Kolmogorov, 1933) test considers the distribution of the largest absolute difference between the empirical and the theoretical cumulative density function across all values.

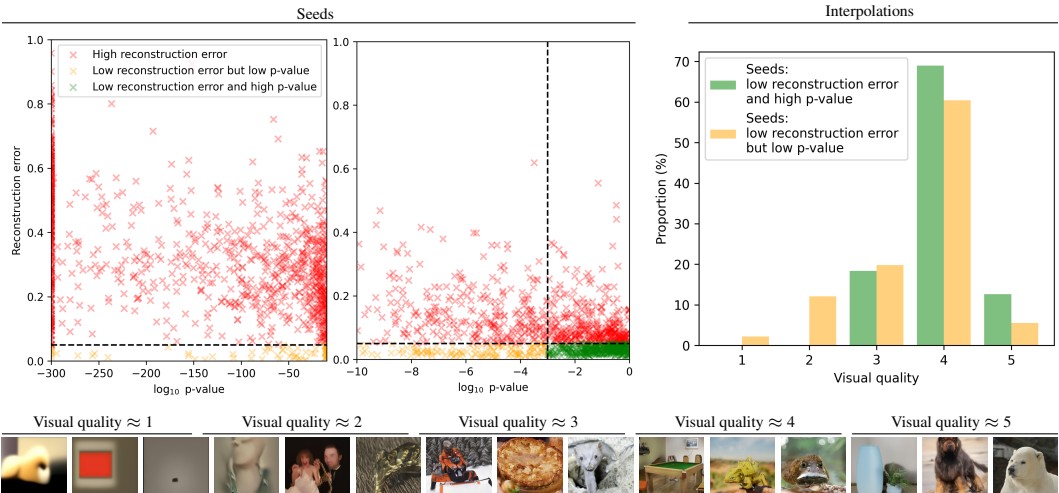

Figure 5: **Lack of normality is linked to failure of interpolants** The left and middle panels shows LPIPS (Zhang et al., 2018) reconstruction error (after 999 generation steps) and the Kolmogorov-Smirnov p-value for all inversions presented in Figure 4, split into two plots due to the vast dynamic range of p-values. Although latents with high (realistic) p-values tend to have low reconstruction errors, there are many latents with low reconstruction errors that also have low p-values. The right panel shows Q-Align visual quality scores (Wu et al., 2023) for spherical interpolants between pairs of inversions selected from matching ImageNet1k (Deng et al., 2009) image classes, demonstrating that choosing seed latents with both low reconstruction error ($< 0.05$) **and** high p-values ($> 1e^{-3}$) allows us to avoid low-quality interpolants that would arise when choosing seeds by reconstruction error alone. For reference, we include examples of interpolants at each visual quality level.

## 3.1 THE SAMPLE CHARACTERISTICS OF SEED LATENTS AFFECT THEIR INTERPOLATIONS

An ability to identify latents that lack the necessary characteristics for good quality generation is critical when manipulating latents that are not acquired by explicitly sampling from the latent distribution. Indeed, as we show below, using such deficient latents as seeds for interpolation can severely affect the quality of interpolants, for example when interpolating between inverted natural images (the most common way to obtain a seed latent, see Section 2). As inversion is subject to a finite computation budget (i.e. finite inversion steps), numerical imprecision, and because the particular data instance may not be drawn from the distribution of the model's training data, the resulting latent vector $x_*$ may not correspond to a sample from the latent distribution. We will now show that distribution testing is helpful for identifying such deficient latents, providing an opportunity to apply corrections or review the examples or inversion setup, before performing interpolation.

Figure 4 demonstrates that inversions having realistic p-values are strongly correlated with them reproducing their original image with good quality. The p-value indicates the probability of observing the latent vector by chance *given* that it is drawn from the latent distribution $\mathcal{N}(\boldsymbol{\mu}, \boldsymbol{\Sigma})$. If this value is extremely low that is a strong indicator that the latent lacks characteristics expected of samples from $\mathcal{N}(\boldsymbol{\mu}, \boldsymbol{\Sigma})$. In the figure we see that at low inversion budgets (with high reconstruction errors) most p-values are $1e^{-50}$ or lower, and then reach values you expect to see by chance for real samples (around $1e^{-3}$) at budgets where the reconstruction errors tend to be small. However, we will show that the p-value provides additional information about the quality of the latents than provided by reconstruction error alone.

We now examine how the p-values of two inversions relate to our ability to interpolate them with good quality. Figure 5 shows that although latents with realistic p-values tend to have low reconstruction errors, there are many latents with low reconstruction errors that have low p-values. Therefore, just because the process of going from object to latent and back reproduces the object, this does not necessarily mean that the resulting latent is well characterised as a sample from $\mathcal{N}(\boldsymbol{\mu}, \boldsymbol{\Sigma})$. The lack of such characteristics can be inherited when manipulating the latent; in Appendix 8 we show that interpolants typically inherit low p-values from their seed latents. Figure 5 also demonstrates that if

we beyond requiring inversions to reconstruct their original objects also require their p-values to be realistic — at levels expected of real samples — we are able to avoid many low quality interpolants downstream, by not using the rejected latents as seeds. In other words, the normality test helps us discover when the latents acquired from inversion lack characteristics of real samples, as this lack prevents them from being reliably used as seeds. Distribution testing the latents can be helpful for debugging the inversion setup — which typically involves solving an ODE — and assessing the settings used. In Appendix E we investigate the effectiveness of a range of other classic normality tests and tests based on the likelihood $\mathcal{N}(\boldsymbol{x}_*; \boldsymbol{\mu}, \boldsymbol{\Sigma})$ as well as the likelihood of the norm statistic. We assess these methods across a suite of test cases including vectors with unlikely characteristics, and for which we know, through failed generation, violate the assumptions of the generative model.

## 4 LINEAR COMBINATIONS OF LATENTS

Now, equipped with the knowledge that matching broad characteristics of a sample from the latent distribution is critical, we propose a simple scheme for forming linear combinations of seed latents — which we now will assume *are* sampled from the latent distribution — to maintain their sample characteristics. We will focus on Gaussian latents in this section, and present an extension to **general latent distributions** in Appendix A. In this section we change the notation slightly, where a seed latent $\boldsymbol{x}_k \in \mathbb{R}^D$ rather than being a realisation of a random variable — a vector of known values — it is a random variable following the latent distribution (here, $\mathcal{N}(\boldsymbol{\mu}, \boldsymbol{\Sigma})$). We assume access to $K$ such seed latent variables $\{\boldsymbol{x}_k\}_{k=1}^K$ and attempt to form new variables following the same distribution.

Let $\boldsymbol{y}$ be a linear combination of the $K$ Gaussian latent variables $\boldsymbol{x}_k \sim \mathcal{N}(\boldsymbol{\mu}, \boldsymbol{\Sigma})$

$$\boldsymbol{y} := \sum_{k=1}^K w_k \boldsymbol{x}_k = \boldsymbol{w}^T \boldsymbol{X}, \tag{1}$$

where $w_k \in \mathbb{R}$, $\boldsymbol{w} = [w_1, w_2, \dots, w_K]$ and $\boldsymbol{X} = [\boldsymbol{x}_1, \boldsymbol{x}_2, \dots, \boldsymbol{x}_K]$. Then we have that $\boldsymbol{y}$ is also a Gaussian random variable, with mean and covariance

$$\boldsymbol{y} \sim \mathcal{N}(\alpha \boldsymbol{\mu}, \beta \boldsymbol{\Sigma}) \qquad \alpha = \sum_{k=1}^K w_k \qquad \beta = \sum_{k=1}^K w_k^2. \tag{2}$$

In other words, $\boldsymbol{y}$ is only distributed as $\mathcal{N}(\boldsymbol{\mu}, \boldsymbol{\Sigma})$ in the specific case where (a) $\alpha \boldsymbol{\mu} = \boldsymbol{\mu}$ and (b) $\beta \boldsymbol{\Sigma} = \boldsymbol{\Sigma}$ — an observation which we now use to explain the empirically observed behaviour of existing interpolation methods. Firstly, for linear interpolation, where $\boldsymbol{w} = [v, 1-v], v \in [0, 1]$, (b) holds only for the endpoints $v = \{0, 1\}$, and so leads to implausible generations for interpolants (as we demonstrate empirically in Figure 11). In contrast, in the popular case of high-dimensional unit Gaussian latent vectors, spherical interpolants have $\beta \approx 1, \forall v \in [0, 1]$, as proven in Appendix C, and (a) is met as $\alpha \boldsymbol{0} = \boldsymbol{0}$, which is consistent with plausible interpolations (see Figure 11).

In this work, we instead propose transforming linear combinations such that $\alpha = \beta = 1$, for any $\boldsymbol{w} \in \mathbb{R}^K$, thus **exactly meeting the criteria** for **any** linear combination and **any** choice of $\boldsymbol{\mu}$ and $\boldsymbol{\Sigma}$. We define a transformed random variable $\boldsymbol{z}$ to use as the latent instead of the linear combination $\boldsymbol{y}$

$$\boldsymbol{z} := \mathcal{T}_{\boldsymbol{w}, \boldsymbol{\mu}}(\boldsymbol{y}) = \mathcal{T}_{\boldsymbol{w}, \boldsymbol{\mu}}(\sum_{k=1}^K w_k \boldsymbol{x}_k), \tag{3}$$

where $\mathcal{T}_{\boldsymbol{w}, \boldsymbol{\mu}}(\boldsymbol{y}) := (1 - \frac{\alpha}{\sqrt{\beta}})\boldsymbol{\mu} + \frac{\boldsymbol{y}}{\sqrt{\beta}}$, for which it holds that $\boldsymbol{z} \sim \mathcal{N}(\boldsymbol{\mu}, \boldsymbol{\Sigma})$ given latent variables $\boldsymbol{x}_k \sim \mathcal{N}(\boldsymbol{\mu}, \boldsymbol{\Sigma})$. Here, $\mathcal{T}_{\boldsymbol{w}, \boldsymbol{\mu}} : \mathbb{R}^D \to \mathbb{R}^D$ is the map that transforms samples from the distribution of a linear combination of $\mathcal{N}(\boldsymbol{\mu}, \boldsymbol{\Sigma})$-variables with weights $\boldsymbol{w}$ into $\mathcal{N}(\boldsymbol{\mu}, \boldsymbol{\Sigma})$. In Appendix B we show that this transport map is Monge optimal, and extend it to general distributions in Appendix A. The weights $\boldsymbol{w}$, which via $\alpha$ and $\beta$ together with the set of $K$ seed latents specify the transformed linear combination $\boldsymbol{z}$, depend on the operation, represented as particular linear combinations. Below are a few examples of popular operations (linear combinations) to form $\boldsymbol{y}$; these are used as above to, for the corresponding weights $\boldsymbol{w}$, obtain $\boldsymbol{z}$ following the distribution expected by the generative model.

- **Interpolation**: $\boldsymbol{y} = \boldsymbol{w}^T \boldsymbol{X}$, where $\boldsymbol{X} = [\boldsymbol{x}_1, \boldsymbol{x}_2]$, $\boldsymbol{w} = [w_1, 1 - w_1]$ and $w_1 \in [0, 1]$.

- **Centroid Determination**: $\boldsymbol{y} = \boldsymbol{w}^T \boldsymbol{X}$, where $\boldsymbol{X} = [\boldsymbol{x}_1, \ldots, \boldsymbol{x}_K], \boldsymbol{w} = [\frac{1}{K}]^K$.

- **Subspaces**: Suppose we wish to build a navigable subspace spanned by linear combinations of $K$ latent variables. By performing the QR decomposition of $\boldsymbol{X} := [\boldsymbol{x}_1, \boldsymbol{x}_2, \ldots, \boldsymbol{x}_K] \in \mathbb{R}^{D \times K}$ to produce a semi-orthonormal matrix $\boldsymbol{U} \in \mathbb{R}^{D \times K}$ (as the Q-matrix), we can then define a subspace projection of any new $\boldsymbol{x}$ into the desired subspace via $s(\boldsymbol{x}) := \boldsymbol{U}\boldsymbol{U}^T\boldsymbol{x} = \boldsymbol{U}\boldsymbol{h} \in \mathbb{R}^D$. The weights $\boldsymbol{w}$ for a given point in the subspace $s(\boldsymbol{x})$ are given by $\boldsymbol{w} = \boldsymbol{X}_\dagger s(\boldsymbol{x}) = \boldsymbol{X}_\dagger \boldsymbol{U}\boldsymbol{h} \in \mathbb{R}^K$ where $\boldsymbol{X}_\dagger$ is the Moore–Penrose inverse of $\boldsymbol{X}$. See the derivation of the weights and proof in Appendix D. One can directly pick coordinates $\boldsymbol{h} \in \mathbb{R}^K$, compute the subspace projection $\boldsymbol{y} = \boldsymbol{U}\boldsymbol{h}$, and then subsequently use the transport map (defined by Equation 3) to the latent space to obtain $\boldsymbol{z}$. In Figure 1 we use grids in $\mathbb{R}^K$ to set $\boldsymbol{h}$, used as above together with a basis (defined by $\boldsymbol{U}$) from a set of latents.

## 5 EXPERIMENTS

We now assess our proposed transformation scheme LOL experimentally. To verify that LOL matches or exceeds current methods for Gaussian latents for currently available operations — interpolation and centroid determination — we perform qualitative and quantitative comparisons to their respective baselines. We then demonstrate new capabilities with several examples of low-dimensional subspaces on popular diffusion models and a popular flow matching model.

### 5.1 INTERPOLATION AND CENTROID DETERMINATION

For the application of interpolation, we compare our proposed LOL to linear interpolation (LERP), spherical linear interpolation (SLERP), and Norm-Aware Optimization (NAO) Samuel et al. (2023).

In contrast to the all the other considered interpolation methods (including LOL) which only involve closed-form expressions, NAO requires numerical optimisation. We closely follow the evaluation protocol in Samuel et al. (2023), basing the experiments on Stable Diffusion (SD) 2.1 (Rombach et al., 2022) and inversions of random images from 50 random classes from ImageNet1k (Deng et al., 2009), and assess visual quality and preservation of semantics using Fréchet Inception Distance (FID) (Heusel et al., 2017) and class prediction accuracy, respectively. For the interpolation we (randomly without replacement) pair the 50 images per class into 25 pairs, forming 1250 image pairs in total.

| Interpolation | | | |
|---|---|---|---|
| **Method** | **Accuracy** | **FID** $\downarrow$ | **Time** |
| LERP | 3.92% | 199 | $6e^{-3}$s |
| SLERP | 64.6% | 42.6 | $9e^{-3}$s |
| NAO | 62.1% | 46.0 | 30s |
| LOL (ours) | 67.4% | 38.9 | $6e^{-3}$s |
| Centroid determination | | | |
| **Method** | **Accuracy** | **FID** $\downarrow$ | **Time** |
| Euclidean | 0.286% | 310 | $4e^{-4}$s |
| Standardised Euclidean | 44.6% | 88.8 | $1e^{-3}$s |
| Mode norm Euclidean | 44.6% | 88.4 | $1e^{-3}$s |
| NAO | 44.0% | 93.0 | 90s |
| LOL (ours) | 46.3% | 87.7 | $6e^{-4}$s |

Table 1: Quantitative comparisons of baselines.

For the centroid determination we compare to NAO, the Euclidean centroid $\bar{\boldsymbol{x}} = \frac{1}{K}\sum_{k=1}^{K} \boldsymbol{x}_k$ and two transformations thereof; "standardised Euclidean", where $\bar{\boldsymbol{x}}$ is subsequently standardised to have mean zero and unit variance (as SD 2.1 assumes), and "mode norm Euclidean", where $\bar{\boldsymbol{x}}$ is rescaled for its norm to equal the maximum likelihood norm $\sqrt{D}$. For each class, we form 10 3-groups, 10 5-groups, 4 10-groups and 1 25-group, sampled without replacement per group, for a total of 1250 centroids per method. For more details on the experiment setup and settings, see Appendix F.

In Table 1 we show that our method outperforms or maintains the performance of the baselines in terms of FID distance and accuracy, as calculated using a pre-trained classifier following the evaluation methodology of Samuel et al. (2023). For an illustration of centroids and interpolations, see Figure 2 and Figure 16, respectively. The evaluation time of an interpolation path and centroid, shown with one digit of precision, illustrate that the closed-form expressions are significantly faster than NAO. Surprisingly, NAO did not perform as well as spherical interpolation and several other baselines, despite using their implementation, which was outperforming these methods in Samuel et al. (2023). We note one discrepancy is that we report FID distances using (2048) high-level features, while in their work they are using (64) low-level features, which in Seitzer (2020) is recommended

against as it does not necessarily correlate with visual quality. In the appendix we include FID distances using all settings of features. We note that, in our setup, the baselines perform substantially better than reported in Samuel et al. (2023) — including NAO in terms of FID distances using the 64 low-level feature setting (1.30 in our setup vs 6.78 in their setup), and class accuracy during interpolation (62% vs 52%). See Section H in the appendix for more details and ablations.

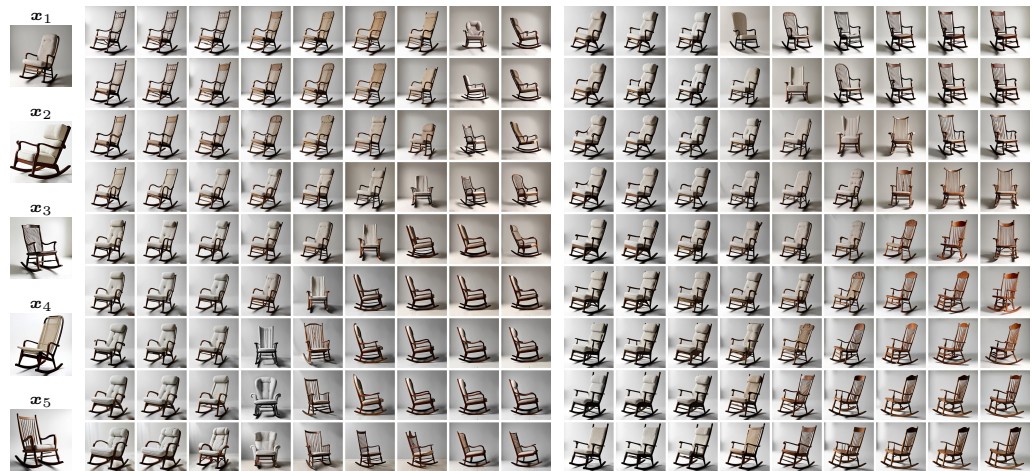

Figure 6: **Low-dimensional subspaces**. The latents $x_1, \ldots, x_5$ (corresponding to images) are converted into basis vectors and used to define a 5-dimensional subspace. The grids show generations from the flow matching model Stable Diffusion 3 (Esser et al., 2024) over uniform grid points in the subspace coordinate system, where the left and right grids are for the dimensions $\{1, 2\}$ and $\{3, 4\}$, respectively, centered around the coordinate for $x_1$. Each coordinate in the subspace correspond to a linear combination of the basis vectors, which through LOL all yield high-quality generations.

## 5.2 LOW-DIMENSIONAL SUBSPACES

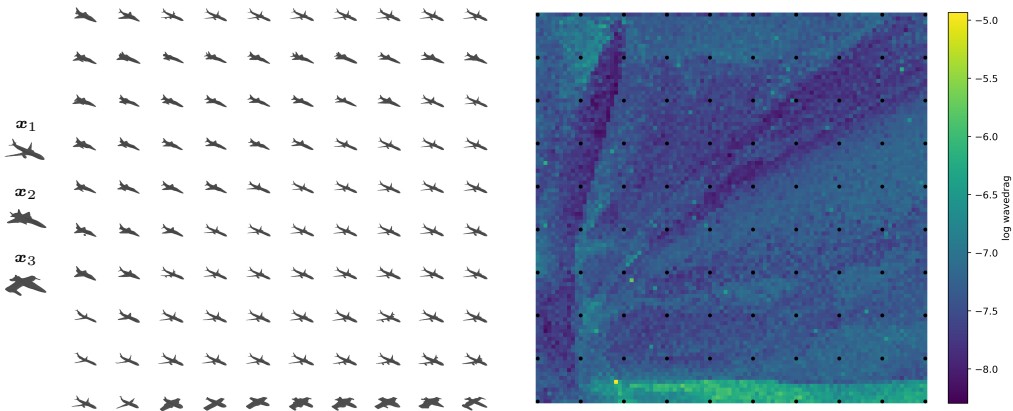

Figure 7: **Model-agnostic subspace definitions**. The latents $x_1, x_2$ and $x_3$ — which via the generative model correspond to 3D designs — are converted into basis vectors and used to define a three-dimensional subspace. The left plot shows the generated designs corresponding to a ten-by-ten grid in a two-dimensional slice of the subspace, shown over a region. The right plot shows the wavedrag, as evaluated by simulation in OpenVSP (McDonald & Gloudemans, 2022) for the designs over a 100-by-100 grid in the same region, with the respective design coordinates (from the left plot) shown as black dots. We trained the SLIDE (Lyu et al., 2023) diffusion model on ShapeNet (Chang et al., 2015). LOL allows subspaces to be defined without any model-specific treatment.

In Figure 1 and Figure 6 we illustrate slices of a 5-dimensional subspace of the latent space of a flow matching model, indexing high-dimensional images of sports cars and rocking chairs, respectively.

The subspaces here are defined using five images (one per desired dimension), formed using our LOL method described in Section 4 to transform the linear combinations of the corresponding projections. Our approach is dimensionality and model-agnostic, which we illustrate in Figure 7, where the same procedure is used on a completely different model without adaptations — where we define a three-dimensional subspace based on three designs in a point cloud diffusion model with mesh reconstruction. We evaluate designs in this subspace using a simple computational fluid dynamics simulation to illustrate that our approach allows objective functions to be defined over the space, in turn allowing off-the-shelf optimisation methods to be applied to search for designs. In Figure 9 in the appendix we show corresponding grids to the sports cars in Figure 1 without the proposed LOL transformation, which either leads to implausible images or the same image for each coordinate, depending on the model. In the appendix (Section G) we also include more slices and examples; including subspaces using the diffusion model Stable Diffusion 2.1 (Rombach et al., 2022).

## 6 RELATED WORK

**Generative models with non-Gaussian priors**. In Fadel et al. (2021) and Davidson et al. (2018), in the context of normalizing flows and VAEs, respectively, Dirichlet and von Mises-Fisher prior distributions are explored with the goal of improving interpolation performance. However, these methods require training the model with the new latent distribution, which is impractical and untested for the large pretrained models we consider.

**Conditional Diffusion**. An additional way to control the generation process of a diffusion model is to guide the generative process with additional information, such as text and labels, to produce outputs that meet specific requirements, such as where samples are guided towards a desired class (Dhariwal & Nichol, 2021; Ho & Salimans, 2022), or to align outputs with text descriptions (Radford et al., 2021). Conditioning is complementary to latent space manipulation. For example, when making Figure 1 we used conditioning (a prompt) to constrain the generation to sports cars, however, the variation of images fulfilling this constraint is encoded in the latent space.

**Low-dimensional representations**. We have shown that LOL can provide expressive low-dimensional representations of latent spaces of generative models. To the best of our knowledge, the most closely related line of work was initiated in Kwon et al. (2022), where it was shown that semantic edit directions can recovered from activations of a UNet (Ronneberger et al., 2015) denoiser network during generation. Using a pretrained CLIP (Contrastive Language-Image Pre-Training) (Radford et al., 2021) model to define directions within the inner-most feature map of the UNet architecture, named h-space, they show that some semantic edits, such as adding glasses to an image of a person, correspond to linear edits in the h-space. More recently, Haas et al. (2024) demonstrate that h-space edit directions can also be found through Principle Component Analysis, and Park et al. (2023) propose a pullback metric that transfers edits in the h-space into the original latent space. However, while these three approaches demonstrate impressive editing capabilities on images, they all require a modification to the generative process and are limited to diffusion models built with UNet architectures. In our work low-dimensional representations are instead formed in closed-form as linear subspaces based on a (free) choice of latent basis vectors. These basis vectors could be obtained through various methodologies, including via the pull-back metric in Park et al. (2023), which may be interesting to explore in future work.

## 7 CONCLUSION

In this paper we propose LOL, a general and simple scheme to define linear combinations of latents that maintain a prespecified latent distribution and demonstrate its effectiveness for interpolation and defining subspaces within the latent spaces of generative models. Of course, these linear combinations only follow the desired distribution if the original (seed) latents do. Therefore, we propose the adoption of distribution tests to assess the validity of latents obtained from e.g. inversions. Still, the existing methods for distribution testing may not align perfectly with a given network's expectations — e.g. many tests may be stricter than necessary for the network — and it may be an interesting direction of future work to develop tailored methods, along with more extensive comparisons of existing tests in the context of generative models.

ACKNOWLEDGMENTS

We thank Samuel Willis and Daattavya Aggarwal (University of Cambridge) for their valuable contributions. Samuel contributed important ideas related to computational fluid dynamics simulation and the SLIDE model, and constructed the setup and plots used in Figure 7. Daattavya assisted in verifying proofs and offered comments that greatly improved the manuscript.

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

## A OPTIMAL TRANSPORT MAP FOR GENERAL LATENT LINEAR COMBINATIONS WITH INDEPENDENT ELEMENTS

In Section 4 we introduced LOL for Gaussian latent variables, the most commonly used latent distribution in modern generative models. We now extend LOL to general latent distributions with independent real-valued elements, provided the cumulative distribution function (CDF) and its inverse is known for each respective element.

Let $p(\boldsymbol{x})$ be a distribution with independent components across elements, such that $p(\boldsymbol{x}) = \prod_{d=1}^{D} p_d(x^{(d)})$, where $p_d$ is the distribution of element $x^{(d)} \in \mathbb{R}$ with CDF $\Phi_d$ and inverse CDF $\Phi_d^{-1}$.

Given linear combination weights $\{w_k\}_{k=1}^K$, $w_k \in \mathbb{R}$, and seeds $\{x_k\}_{k=1}^K$, $x_k \sim p$, we obtain $z \sim p$ via the transport map $\mathcal{T}_{\{w\}} := \{\mathcal{T}_{\{w\}}^{(d)}\}_{d=1}^D$ defined per latent element $z^{(d)}$ as:

$$z^{(d)} := \mathcal{T}_{\{w\}}^{(d)}(\{x_k^{(d)}\}) = \Phi_d^{-1}(\Phi_\epsilon(\epsilon^{(d)})) \quad \epsilon^{(d)} = \mathcal{T}_{w,0}\left(\sum_{k=1}^K w_k \epsilon_k^{(d)}\right) \quad \epsilon_k^{(d)} = \Phi_\epsilon^{-1}(\Phi_d(x_k^{(d)})), \quad (4)$$

where $\Phi_\epsilon$ is the CDF of the standard Gaussian, and $\mathcal{T}_{w,0} : \mathbb{R} \to \mathbb{R}$ is the transport map for standard Gaussians as defined in Equation 3. We will now show that the proposed map leads to the target distribution $p$ and then show that this map is Monge optimal, leveraging results from Section B.

**Lemma 1.** *Let $z$ be a transformed variable defined through the transport map $\mathcal{T}_{\{w\}}$ applied to a linear combination of independent latent variables $x_k \sim p(x)$, such that*

$$z^{(d)} = \mathcal{T}_{\{w\}}^{(d)}(\{x_k^{(d)}\}) = \Phi_d^{-1}(\Phi_\epsilon(\mathcal{T}_{w,0}(\sum_{k=1}^K w_k \epsilon_k^{(d)})))$$

*where*

$$\epsilon_k^{(d)} = \Phi_\epsilon^{-1}(\Phi_d(x_k^{(d)})).$$

*Then, $z \sim p(x)$, meaning that the transformed variable follows the target latent distribution.*

*Proof.* We prove that $z \sim p(x)$ by verifying that each element $z^{(d)}$ follows the corresponding marginal distribution $p_d$.

Since $x_k^{(d)} \sim p_d$, its cumulative distribution function (CDF) is given by $\Phi_d(x_k^{(d)})$. Defining the standard normal equivalent $\epsilon_k^{(d)} = \Phi_\epsilon^{-1}(\Phi_d(x_k^{(d)}))$, we know that $\epsilon_k^{(d)} \sim \mathcal{N}(0,1)$ because the transformation via $\Phi_\epsilon^{-1}$ preserves uniformity.

Now, consider the weighted sum:

$$\epsilon^{(d)} = \sum_{k=1}^K w_k \epsilon_k^{(d)}.$$

Since the $\epsilon_k^{(d)}$ are independent standard normal variables, the resulting sum follows:

$$\epsilon^{(d)} \sim \mathcal{N}(\alpha \cdot 0, \beta \cdot 1) = \mathcal{N}(0, \beta),$$

where $\alpha = \sum_{k=1}^K w_k$ and $\beta = \sum_{k=1}^K w_k^2$.

Applying the optimal Gaussian transport map $\mathcal{T}_{w,0}$ (as proven in Section B) to $\epsilon^{(d)}$, we obtain:

$$\mathcal{T}_{w,0}(\epsilon^{(d)}) = \frac{\epsilon^{(d)}}{\sqrt{\beta}}.$$

Since $\epsilon^{(d)} \sim \mathcal{N}(0, \beta)$, it follows that:

$$\frac{\epsilon^{(d)}}{\sqrt{\beta}} \sim \mathcal{N}(0, 1).$$

Thus, applying the standard normal CDF, we get:

$$\Phi_\epsilon(\mathcal{T}_{w,0}(\epsilon^{(d)})) \sim U(0, 1),$$

which is a uniform distribution.

Finally, applying the inverse CDF $\Phi_d^{-1}$ of the target distribution $p_d$ results in:

$$z^{(d)} = \Phi_d^{-1}(\Phi_\epsilon(\mathcal{T}_{w,0}(\epsilon^{(d)}))) \sim p_d.$$

Since this holds for each $d$, we conclude that $z \sim p(x)$, proving that the transport map correctly transforms the linear combination into the desired latent distribution. $\square$

**Lemma 2.** *The transport map $\mathcal{T}_{\{w\}}$ defined element-wise as*

$$z^{(d)} = \mathcal{T}_{\{w\}}^{(d)}(\{x_k^{(d)}\}) = \Phi_d^{-1}(\Phi_\epsilon(\mathcal{T}_{\boldsymbol{w},0}(\sum_{k=1}^{K} w_k \epsilon_k^{(d)})))$$

*where*

$$\epsilon_k^{(d)} = \Phi_\epsilon^{-1}(\Phi_d(x_k^{(d)})),$$

*is the Monge optimal transport map, minimizing the quadratic Wasserstein distance $W_2$ between the source distribution of the weighted sum of independent elements and the target latent distribution $p$.*

*Proof.* We want to show that $\mathcal{T}_{\{w\}}$ minimizes the quadratic Wasserstein distance $W_2$ between the source distribution (the weighted sum of independent seeds) and the target distribution.

**Step 1: General Theorem for One-Dimensional Optimal Transport** For two probability measures $P$ and $Q$ on $\mathbb{R}$ with cumulative distribution functions (CDFs) $F_P$ and $F_Q$, respectively, the optimal transport map in the $W_2$ sense (which minimizes the expected squared Euclidean distance) is given by the monotone increasing function:

$$\mathcal{T}^* = F_Q^{-1} \circ F_P.$$

This result follows from classical optimal transport theory for univariate distributions (see Remark 2.30 in Peyré & Cuturi (2020)).

**Step 2: Applying to Our Case** For each element $d$, the source distribution is given by the weighted sum:

$$\epsilon^{(d)} = \sum_{k=1}^{K} w_k \epsilon_k^{(d)}.$$

Each $\epsilon_k^{(d)}$ is obtained by transforming the original variable $x_k^{(d)}$ using:

$$\epsilon_k^{(d)} = \Phi_\epsilon^{-1}(\Phi_d(x_k^{(d)})).$$

Since the $\epsilon_k^{(d)}$ are independent standard normal variables, their weighted sum follows:

$$\epsilon^{(d)} \sim \mathcal{N}(0, \beta), \quad \text{where} \quad \beta = \sum_{k=1}^{K} w_k^2.$$

To map this distribution to the standard normal $\mathcal{N}(0, 1)$, the optimal transport map for Gaussians (derived in Section B) is:

$$\mathcal{T}_{\boldsymbol{w},0}(\epsilon^{(d)}) = \frac{\epsilon^{(d)}}{\sqrt{\beta}}.$$

Applying the standard normal CDF $\Phi_\epsilon$ to this transformation ensures that:

$$\Phi_\epsilon(\mathcal{T}_{\boldsymbol{w},0}(\epsilon^{(d)})) \sim U(0, 1).$$

By the general optimal transport theorem stated in Step 1, the optimal transport map to the target distribution $p_d$ is then:

$$z^{(d)} = \Phi_d^{-1}(\Phi_\epsilon(\mathcal{T}_{\boldsymbol{w},0}(\epsilon^{(d)}))).$$

Since each element $d$ is mapped independently using the element-wise Monge optimal transport map, the full transformation $\mathcal{T}_{\{w\}}$ is optimal in the Monge sense, minimizing the quadratic Wasserstein distance $W_2$.

Thus, $\mathcal{T}_{\{w\}}$ is the Monge optimal transport map from the distribution of the weighted sum of independent elements to the target latent distribution $p$, completing the proof. $\qquad\square$

## B    Optimal transport map for Gaussian latent linear combinations

In Section 4 we introduced LOL, which we use to transform random variables arising from linear combinations — for example interpolations or subspace projections — to the latent distribution $\mathcal{N}(\boldsymbol{\mu}, \boldsymbol{\Sigma})$ via a transport map $\mathcal{T}_{\boldsymbol{w},\boldsymbol{\mu}}$ defined per set of linear combination weights $\boldsymbol{w} = [w_1, \ldots, w_K], w_k \in \mathbb{R}$. We will now derive the distribution of a (uncorrected) linear combination $\boldsymbol{y}$ — which we show does not match the latent distribution — followed by the distribution of the transformed variable $\boldsymbol{z}$, which we show does. We will then show that the proposed transport map $\mathcal{T}_{\boldsymbol{w},\boldsymbol{\mu}}$ is Monge optimal.

**Lemma 3.** *Let $\boldsymbol{y}$ be a linear combination of $K$ i.i.d. random variables $\boldsymbol{x}_k \sim \mathcal{N}(\boldsymbol{\mu}, \boldsymbol{\Sigma})$, where $\boldsymbol{y}$ is defined as*

$$\boldsymbol{y} := \sum_{k=1}^{K} w_k \boldsymbol{x}_k = \boldsymbol{w}^T \boldsymbol{X},$$

*with $w_k \in \mathbb{R}$, $\boldsymbol{y} \in \mathbb{R}^D$, $\boldsymbol{x}_k \in \mathbb{R}^D$, $\boldsymbol{w} = [w_1, w_2, \ldots, w_K]$, $K \in \mathbb{N}_{>0}$ and $\boldsymbol{X} = [\boldsymbol{x}_1, \boldsymbol{x}_2, \ldots, \boldsymbol{x}_K]$. Then $\boldsymbol{y}$ is a Gaussian random variable with the distribution*

$$\boldsymbol{y} \sim \mathcal{N}(\alpha \boldsymbol{\mu}, \beta \boldsymbol{\Sigma}),$$

*where*

$$\alpha = \sum_{k=1}^{K} w_k \quad \text{and} \quad \beta = \sum_{k=1}^{K} w_k^2.$$

*Proof.* Given $\boldsymbol{x}_k \sim \mathcal{N}(\boldsymbol{\mu}, \boldsymbol{\Sigma})$, we know that each $\boldsymbol{x}_k$ has mean $\boldsymbol{\mu}$ and covariance matrix $\boldsymbol{\Sigma}$.

Define $\boldsymbol{y} = \boldsymbol{w}^T \boldsymbol{X} = \sum_{k=1}^{K} w_k \boldsymbol{x}_k$.

First, we calculate the mean of $\boldsymbol{y}$:

$$\mathbb{E}[\boldsymbol{y}] = \mathbb{E}\left[\sum_{k=1}^{K} w_k \boldsymbol{x}_k\right] = \sum_{k=1}^{K} w_k \mathbb{E}[\boldsymbol{x}_k] = \sum_{k=1}^{K} w_k \boldsymbol{\mu} = \left(\sum_{k=1}^{K} w_k\right) \boldsymbol{\mu} = \alpha \boldsymbol{\mu}.$$

Next, we calculate the covariance of $\boldsymbol{y}$:

$$\text{Cov}(\boldsymbol{y}) = \text{Cov}\left(\sum_{k=1}^{K} w_k \boldsymbol{x}_k\right) = \sum_{k=1}^{K} w_k^2 \text{Cov}(\boldsymbol{x}_k),$$

since the $\boldsymbol{x}_k$ are i.i.d. and thus $\text{Cov}(\boldsymbol{x}^{(i)}, \boldsymbol{x}^{(j)}) = 0$ for $i \neq j$.

Given that $\boldsymbol{x}_k \sim \mathcal{N}(\boldsymbol{\mu}, \boldsymbol{\Sigma})$, we have $\text{Cov}(\boldsymbol{x}_k) = \boldsymbol{\Sigma}$. Therefore,

$$\text{Cov}(\boldsymbol{y}) = \sum_{k=1}^{K} w_k^2 \boldsymbol{\Sigma} = \left(\sum_{k=1}^{K} w_k^2\right) \boldsymbol{\Sigma} = \beta \boldsymbol{\Sigma}.$$

Hence, $\boldsymbol{y} \sim \mathcal{N}(\alpha \boldsymbol{\mu}, \beta \boldsymbol{\Sigma})$ with $\alpha = \sum_{k=1}^{K} w_k$ and $\beta = \sum_{k=1}^{K} w_k^2$.

$\qquad\square$

**Lemma 4.** *Let $\boldsymbol{z}$ be defined as*

$$\boldsymbol{z} = \mathcal{T}_{\boldsymbol{w},\boldsymbol{\mu}}(\boldsymbol{y}) \qquad \mathcal{T}_{\boldsymbol{w},\boldsymbol{\mu}}(\boldsymbol{y}) = (1 - \frac{\alpha}{\sqrt{\beta}})\boldsymbol{\mu} + \frac{\boldsymbol{y}}{\sqrt{\beta}} \qquad \alpha = \sum_{k=1}^{K} w_k \qquad \beta = \sum_{k=1}^{K} w_k^2$$

*with $\boldsymbol{y} \sim \mathcal{N}(\alpha\boldsymbol{\mu}, \beta\boldsymbol{\Sigma})$. Then $\boldsymbol{z} \sim \mathcal{N}(\boldsymbol{\mu}, \boldsymbol{\Sigma})$.*

*Proof.* Given $\boldsymbol{y} \sim \mathcal{N}(\alpha\boldsymbol{\mu}, \beta\boldsymbol{\Sigma})$, we need to show that $\boldsymbol{z} \sim \mathcal{N}(\boldsymbol{\mu}, \boldsymbol{\Sigma})$.

First, we calculate the mean of $\boldsymbol{z}$:

$$\mathbb{E}[\boldsymbol{z}] = \mathbb{E}\left[\boldsymbol{a} + \boldsymbol{B}\boldsymbol{y}\right] = \boldsymbol{a} + \boldsymbol{B}\mathbb{E}[\boldsymbol{y}] = \left(1 - \frac{\alpha}{\sqrt{\beta}}\right)\boldsymbol{\mu} + \frac{1}{\sqrt{\beta}}\alpha\boldsymbol{\mu} = \boldsymbol{\mu},$$

where

$$\boldsymbol{a} = \left(1 - \frac{\alpha}{\sqrt{\beta}}\right)\boldsymbol{\mu} \quad \text{and} \quad \boldsymbol{B} = \frac{1}{\sqrt{\beta}}\boldsymbol{I}.$$

Next, we calculate the covariance of $\boldsymbol{z}$:

$$\text{Cov}(\boldsymbol{z}) = \text{Cov}\left(\boldsymbol{a} + \boldsymbol{B}\boldsymbol{y}\right) = \boldsymbol{B}\text{Cov}(\boldsymbol{y})\boldsymbol{B}^T = \frac{1}{\sqrt{\beta}}\beta\boldsymbol{\Sigma}\frac{1}{\sqrt{\beta}} = \boldsymbol{\Sigma}.$$

Since $\boldsymbol{z}$ has mean $\boldsymbol{\mu}$ and covariance $\boldsymbol{\Sigma}$, we conclude that

$$\boldsymbol{z} \sim \mathcal{N}(\boldsymbol{\mu}, \boldsymbol{\Sigma}).$$

$\square$

**Lemma 5.** *The transport map $\mathcal{T}_{\boldsymbol{w},\boldsymbol{\mu}}$ defined by*

$$\mathcal{T}_{\boldsymbol{w},\boldsymbol{\mu}}(\boldsymbol{y}) = (1 - \frac{\alpha}{\sqrt{\beta}})\boldsymbol{\mu} + \frac{\boldsymbol{y}}{\sqrt{\beta}}$$

*is the Monge optimal transport map, minimizing the quadratic Wasserstein distance $W_2$ between $\mathcal{N}(\alpha\boldsymbol{\mu}, \beta\boldsymbol{\Sigma})$ and $\mathcal{N}(\boldsymbol{\mu}, \boldsymbol{\Sigma})$.*

*Proof.* In optimal transport theory, the Monge problem seeks a transport map $\mathcal{T} : \mathbb{R}^D \to \mathbb{R}^D$ that pushes forward the source distribution $\mathcal{N}(\boldsymbol{\mu}_y, \boldsymbol{\Sigma}_y)$ to the target distribution $\mathcal{N}(\boldsymbol{\mu}_z, \boldsymbol{\Sigma}_z)$ while minimizing the expected squared Euclidean distance:

$$W_2^2(\mathcal{N}(\boldsymbol{\mu}_y, \boldsymbol{\Sigma}_y), \mathcal{N}(\boldsymbol{\mu}_z, \boldsymbol{\Sigma}_z)) = \inf_{\mathcal{T} \in \mathcal{M}} \mathbb{E}[\|\mathcal{T}(\boldsymbol{y}) - \boldsymbol{y}\|^2].$$

The optimal transport map for Gaussian distributions is given by the affine transformation:

$$\mathcal{T}(\boldsymbol{y}) = \boldsymbol{\mu}_z + \boldsymbol{A}(\boldsymbol{y} - \boldsymbol{\mu}_y),$$

where $\boldsymbol{A}$ is defined as

$$\boldsymbol{A} := \boldsymbol{\Sigma}_y^{-1/2}(\boldsymbol{\Sigma}_y^{1/2}\boldsymbol{\Sigma}_z\boldsymbol{\Sigma}_y^{1/2})^{1/2}\boldsymbol{\Sigma}_y^{-1/2}.$$

See Remark 2.31 in Peyré & Cuturi (2020).

Setting the parameters

$$\boldsymbol{\mu}_y = \alpha\boldsymbol{\mu}, \qquad\qquad \boldsymbol{\Sigma}_y = \beta\boldsymbol{\Sigma}, \tag{5}$$
$$\boldsymbol{\mu}_z = \boldsymbol{\mu}, \qquad\qquad \boldsymbol{\Sigma}_z = \boldsymbol{\Sigma}, \tag{6}$$

we compute $\boldsymbol{A}$:

$$\boldsymbol{A} = (\beta\boldsymbol{\Sigma})^{-1/2}((\beta\boldsymbol{\Sigma})^{1/2}\boldsymbol{\Sigma}(\beta\boldsymbol{\Sigma})^{1/2})^{1/2}(\beta\boldsymbol{\Sigma})^{-1/2} = \frac{1}{\sqrt{\beta}}\boldsymbol{I}. \tag{7}$$

Thus, the optimal transport map simplifies to

$$\boldsymbol{z} := \mathcal{T}(\boldsymbol{y}) = \boldsymbol{\mu} + \frac{1}{\sqrt{\beta}}(\boldsymbol{y} - \alpha\boldsymbol{\mu}) = (1 - \frac{\alpha}{\sqrt{\beta}})\boldsymbol{\mu} + \frac{\boldsymbol{y}}{\sqrt{\beta}}. \tag{8}$$

Since this matches exactly with $\mathcal{T}_{\boldsymbol{w},\boldsymbol{\mu}}$, we conclude that $\mathcal{T}_{\boldsymbol{w},\boldsymbol{\mu}}$ is the Monge optimal transport map minimising the quadratic Wasserstein distance.

$\square$

## C  SPHERICAL INTERPOLANTS OF HIGH-DIMENSIONAL UNIT GAUSSIAN RANDOM VARIABLES ARE APPROXIMATELY UNIT GAUSSIAN

Spherical linear interpolation (SLERP) (Shoemake, 1985) is defined as

$$\boldsymbol{y} = w_1\boldsymbol{x}_1 + w_2\boldsymbol{x}_2 \tag{9}$$

where

$$w_i := \frac{\sin v_i\theta}{\sin\theta}, \tag{10}$$

$v_i \in [0, 1]$ and $v_2 = 1 - v_1$ and $\cos\theta = \frac{\langle\boldsymbol{x}_1,\boldsymbol{x}_2\rangle}{||\boldsymbol{x}_1||||\boldsymbol{x}_2||}$, where $\cos\theta$ is typically referred to as the *cosine similarity* of $\boldsymbol{x}_1$ and $\boldsymbol{x}_2$.

As such, using Equation 2, we obtain

$$\alpha = \sum_{k=1}^{K} w_k = \frac{\sin v_1\theta}{\sin\theta} + \frac{\sin v_2\theta}{\sin\theta} \quad \text{and} \quad \beta = \frac{\sin^2 v_1\theta}{\sin^2\theta} + \frac{\sin^2 v_2\theta}{\sin^2\theta} \tag{11}$$

As discussed in Section 4, for a linear combination $\boldsymbol{y}$ to be a random variable following distribution $\mathcal{N}(\boldsymbol{\mu}, \boldsymbol{\Sigma})$, given that $\boldsymbol{x}_1$ and $\boldsymbol{x}_2$ do, it must be true that $\alpha\boldsymbol{\mu} = \boldsymbol{\mu}$ and $\beta\boldsymbol{\Sigma} = \boldsymbol{\Sigma}$.

A common case is using unit Gaussian latents (as in e.g. the models Esser et al. (2024) and Rombach et al. (2022) used in this paper), i.e. where $\boldsymbol{\mu} = \boldsymbol{0}, \boldsymbol{\Sigma} = \boldsymbol{I}$. In this case it trivially follows that $\alpha\boldsymbol{\mu} = \boldsymbol{\mu}$ since $\alpha\boldsymbol{0} = \boldsymbol{0}$. We will now show that $\beta \approx 1$ in this special (i.e. unit Gaussian) case.

**Lemma 6.** *Let*

$$\beta = \frac{\sin^2 v\theta}{\sin^2\theta} + \frac{\sin^2(1-v)\theta}{\sin^2\theta},$$

*where* $\cos\theta = \frac{\langle\boldsymbol{x}_1,\boldsymbol{x}_2\rangle}{\|\boldsymbol{x}_1\|\|\boldsymbol{x}_2\|}$ *and* $\boldsymbol{x}_1, \boldsymbol{x}_2 \sim \mathcal{N}(\boldsymbol{0}, \boldsymbol{I})$. *Then* $\beta \approx 1$ *for large* $D$, $\forall v \in [0, 1]$.

*Proof.* Since $\boldsymbol{x}_1, \boldsymbol{x}_2 \sim \mathcal{N}(\boldsymbol{0}, \boldsymbol{I})$, each component $x_{1j}$ and $x_{2j}$ for $j = 1, \ldots, D$ are independent standard normal random variables. The inner product $\langle\boldsymbol{x}_1, \boldsymbol{x}_2\rangle$ is given by:

$$\langle\boldsymbol{x}_1, \boldsymbol{x}_2\rangle = \sum_{j=1}^{D} x_{1j}x_{2j}.$$

The product $x_{1j}x_{2j}$ follows a distribution known as the standard normal product distribution. For large $D$, the sum of these products is approximately normal due to the Central Limit Theorem (CLT), with:

$$\langle\boldsymbol{x}_1, \boldsymbol{x}_2\rangle \sim \mathcal{N}(0, D).$$

Next, consider the norms $\|\boldsymbol{x}_1\|$ and $\|\boldsymbol{x}_2\|$. Each $\|\boldsymbol{x}_i\|^2 = \sum_{j=1}^{D} x_{ij}^2$ is a chi-squared random variable with $D$ degrees of freedom. For large $D$, by the central limit theorem, $\|\boldsymbol{x}_i\|^2 \sim \mathcal{N}(D, 2D)$, and therefore $\|\boldsymbol{x}_i\|$ is approximately $\sqrt{D}$.

Thus, for large $D$,

$$\cos(\theta) = \frac{\langle \boldsymbol{x}_1, \boldsymbol{x}_2 \rangle}{\|\boldsymbol{x}_1\|\|\boldsymbol{x}_2\|} \approx \frac{\mathcal{N}(0, D)}{\sqrt{D} \cdot \sqrt{D}} = \frac{\mathcal{N}(0, D)}{D} = \mathcal{N}\left(0, \frac{1}{D}\right).$$

Thus, $\theta \approx \pi/2$, which implies $\sin(\theta) \approx 1$. Therefore:

$$\beta = \frac{\sin^2(v\theta)}{\sin^2 \theta} + \frac{\sin^2((1-v)\theta)}{\sin^2 \theta} \approx \sin^2(v\theta) + \sin^2((1-v)\theta).$$

Using the identity $\sin^2(a) + \sin^2(b) = 1 - \cos^2(a - b)$,

$$\beta \approx 1 - \cos^2(v\theta - (1-v)\theta) = 1 - \cos^2(\theta)$$

Given $v \in [0, 1]$ and $\theta \approx \pi/2$ for large $D$, the argument of $\cos$ remains small, leading to $\cos(\cdot) \approx 0$. Hence,

$$\beta \approx 1.$$

Therefore, for large $D$, it follows that $\beta \approx 1$.

$\square$

**Empirical verification** To confirm the effectiveness of the approximations used above for values of $D$ which are typical for popular generative models that use unit Gaussian latents, we estimate the confidence interval of $\beta$ using 10k samples of $\boldsymbol{x}_1$ and $\boldsymbol{x}_2$, respectively, where $\boldsymbol{x}_i \sim \mathcal{N}(\boldsymbol{0}, \boldsymbol{I})$. For Stable Diffusion 3 (Esser et al., 2024), a flow matching model with $D = 147456$, the estimated 99% confidence interval of $\beta$ is $[0.9934, 1.0067]$ for $v = 0.5$ where the error is largest. For Stable Diffusion 2.1 (Rombach et al., 2022), a diffusion model with $D = 36864$, the corresponding confidence interval of $\beta$ is $[0.9868, 1.014]$.

## D LINEAR COMBINATION WEIGHTS OF SUBSPACE PROJECTIONS

In Section 4 we introduced LOL, which we use to transform latent variables arising from linear combinations such that they maintain the the latent distribution. This transformation depends on the weights $\boldsymbol{w}$ which specify the linear combination. We will now derive the linear combination weights for subspace projections.

**Lemma 7.** *Let $\boldsymbol{U} \in \mathbb{R}^{D \times K}$ be a semi-orthonormal matrix. For a given point $\boldsymbol{x} \in \mathbb{R}^D$, the subspace projection is $s(\boldsymbol{x}) = \boldsymbol{U}\boldsymbol{U}^T\boldsymbol{x}$. The weights $\boldsymbol{w} \in \mathbb{R}^K$ such that $s(\boldsymbol{x})$ is a linear combination of $\boldsymbol{x}_1, \boldsymbol{x}_2, \ldots, \boldsymbol{x}_K$ (columns of $\boldsymbol{A} = [\boldsymbol{x}_1, \boldsymbol{x}_2, \ldots, \boldsymbol{x}_K] \in \mathbb{R}^{D \times K}$) can be expressed as $\boldsymbol{w} = \boldsymbol{A}_\dagger s(\boldsymbol{x})$, where $\boldsymbol{A}_\dagger$ is the Moore-Penrose inverse of $\boldsymbol{A}$.*

*Proof.* The subspace projection $s(\boldsymbol{x})$ of $\boldsymbol{x} \in \mathbb{R}^D$ is defined as:

$$s(\boldsymbol{x}) = \boldsymbol{U}\boldsymbol{U}^T\boldsymbol{x}.$$

We aim to express $s(\boldsymbol{x})$ as a linear combination of the columns of $\boldsymbol{A} = [\boldsymbol{x}_1, \boldsymbol{x}_2, \ldots, \boldsymbol{x}_K] \in \mathbb{R}^{D \times K}$. That is, we seek $\boldsymbol{w} \in \mathbb{R}^K$ such that:

$$s(\boldsymbol{x}) = \boldsymbol{A}\boldsymbol{w}.$$

By definition, the Moore-Penrose inverse $\boldsymbol{A}_\dagger$ of $\boldsymbol{A}$ satisfies the following properties:

1. $\boldsymbol{A}\boldsymbol{A}_\dagger\boldsymbol{A} = \boldsymbol{A}$

2. $\boldsymbol{A}_\dagger\boldsymbol{A}\boldsymbol{A}_\dagger = \boldsymbol{A}_\dagger$

3. $(\boldsymbol{A}\boldsymbol{A}_\dagger)^T = \boldsymbol{A}\boldsymbol{A}_\dagger$

4. $(\boldsymbol{A}_\dagger\boldsymbol{A})^T = \boldsymbol{A}_\dagger\boldsymbol{A}$

Since $s(\boldsymbol{x})$ is in the subspace spanned by the columns of $\boldsymbol{A}$, there exists a $\boldsymbol{w}$ such that:

$$s(\boldsymbol{x}) = \boldsymbol{A}\boldsymbol{w}.$$

Consider a $\boldsymbol{w}' \in \mathbb{R}^K$ constructed using the Moore-Penrose inverse $\boldsymbol{A}_\dagger$:

$$\boldsymbol{w}' = \boldsymbol{A}_\dagger s(\boldsymbol{x}).$$

We now verify that this $\boldsymbol{w}'$ satisfies the required equation. Substituting back

$$\boldsymbol{A}\boldsymbol{w}' = \boldsymbol{A}(\boldsymbol{A}_\dagger s(\boldsymbol{x}))$$

and using the property of the Moore-Penrose inverse $\boldsymbol{A}\boldsymbol{A}_\dagger\boldsymbol{A} = \boldsymbol{A}$, we get:

$$\boldsymbol{A}\boldsymbol{A}_\dagger s(\boldsymbol{x}) = s(\boldsymbol{x}).$$

Thus:

$$\boldsymbol{A}\boldsymbol{w}' = s(\boldsymbol{x}),$$

which shows that $\boldsymbol{w}' = \boldsymbol{A}_\dagger s(\boldsymbol{x})$ is indeed the correct expression for the weights.

From uniqueness of $\boldsymbol{w}$ for a given set of columns $\boldsymbol{x}_1, \boldsymbol{x}_2, \ldots, \boldsymbol{x}_K$ (see Lemma 8), we have proven that the weights $\boldsymbol{w}$ for a given point in the subspace $s(\boldsymbol{x})$ are given by:

$$\boldsymbol{w} = \boldsymbol{A}_\dagger s(\boldsymbol{x}).$$

$\square$

**Lemma 8.** *The weights $\boldsymbol{w} \in \mathbb{R}^K$ such that $s(\boldsymbol{x})$ is a linear combination of $\boldsymbol{x}_1, \boldsymbol{x}_2, \ldots, \boldsymbol{x}_K$ (columns of $\boldsymbol{A} = [\boldsymbol{x}_1, \boldsymbol{x}_2, \ldots, \boldsymbol{x}_K] \in \mathbb{R}^{D \times K}$) are unique.*

*Proof.* Suppose there exist two different weight vectors $\boldsymbol{w}_1$ and $\boldsymbol{w}_2$ such that both satisfy the equation:

$$s(\boldsymbol{x}) = \boldsymbol{A}\boldsymbol{w}_1 = \boldsymbol{A}\boldsymbol{w}_2.$$

Then, subtracting these two equations gives:

$$\boldsymbol{A}\boldsymbol{w}_1 - \boldsymbol{A}\boldsymbol{w}_2 = \boldsymbol{0}.$$

This simplifies to:

$$\boldsymbol{A}(\boldsymbol{w}_1 - \boldsymbol{w}_2) = \boldsymbol{0}.$$

Let $\boldsymbol{v} = \boldsymbol{w}_1 - \boldsymbol{w}_2$. Then:

$$\boldsymbol{A}\boldsymbol{v} = \boldsymbol{0}.$$

Since $\boldsymbol{v} \in \mathbb{R}^K$, this equation implies that $\boldsymbol{v}$ lies in the null space of $\boldsymbol{A}$. However, the assumption that $\boldsymbol{A}$ has full column rank (since $\boldsymbol{A}$ is used to represent a linear combination for $s(\boldsymbol{x})$) implies that $\boldsymbol{A}$ has no non-zero vector in its null space, i.e., $\boldsymbol{A}\boldsymbol{v} = \boldsymbol{0}$ only when $\boldsymbol{v} = \boldsymbol{0}$.

Therefore:

$$\boldsymbol{v} = \boldsymbol{0} \implies \boldsymbol{w}_1 = \boldsymbol{w}_2.$$

This shows that the weights $\boldsymbol{w}$ are unique, and there cannot be two distinct sets of weights $\boldsymbol{w}_1$ and $\boldsymbol{w}_2$ that satisfy the equation $s(\boldsymbol{x}) = \boldsymbol{A}\boldsymbol{w}$.

Hence, we conclude that the weights $\boldsymbol{w}$ such that $s(\boldsymbol{x})$ is a linear combination of $\boldsymbol{x}_1, \boldsymbol{x}_2, \ldots, \boldsymbol{x}_K$ are unique. $\square$

## E NORMALITY TEST COMPARISON

We will now investigate the effectiveness of a range of classic normality tests and tests based on the likelihood $\mathcal{N}(\boldsymbol{x}_*; \boldsymbol{\mu}, \boldsymbol{\Sigma})$ as well as the likelihood of the norm statistic. We assess these methods across a suite of test cases including for vectors with unlikely characteristics and which we know, through failed generation, violate the assumptions of the model. In order to improve statistical power and detect even small deviations from normality of a single instance $\boldsymbol{x}_* \in \mathbb{R}^D$, we transform the single high-dimensional latent into a large collection of $(D)$ i.i.d. unit Gaussian samples $\{\epsilon_i\}_{i=1}^D$, and test the hypothesis $\epsilon_1, \ldots, \epsilon_d \sim \mathcal{N}(0,1)$ where $\boldsymbol{\epsilon} = \boldsymbol{\Sigma}^{-1}(\boldsymbol{x} - \boldsymbol{\mu}), \boldsymbol{\epsilon} = [\epsilon_1, \ldots, \epsilon_d]$. Note that rejecting the hypothesis $\epsilon_1, \ldots, \epsilon_d \sim \mathcal{N}(0,1)$ corresponds to rejecting the hypothesis that $\boldsymbol{x}_* \sim \mathcal{N}(\boldsymbol{\mu}, \boldsymbol{\Sigma})$.

We compare popular alternatives for normality testing that assume a known mean and variance and that are applicable to very large sample sizes, including Kolmogorov-Smirnov (Kolmogorov, 1933), Shapiro-Wilk (Shapiro & Wilk, 1965), Chi-square (Pearson, 1900), and Cramér–von Mises (Cramér, 1928). We assess these methods on a suite of test cases of vectors which are extremely unlikely under the model's latent distribution and which we know through failed generation violates the characteristic assumptions of the diffusion model, as well as assess the "positive case" where the sample *does* follow the latent distribution, $\boldsymbol{x}_* \sim \mathcal{N}(\boldsymbol{\mu}, \boldsymbol{\Sigma})$. We report the $[0.1\%, 99.9\%]$-confidence interval of the p-value produced by each respective method on $1e^4$ random samples in the stochastic test cases. We also include the likelihood $\mathcal{N}(\boldsymbol{x}_*; \boldsymbol{\mu}, \boldsymbol{\Sigma})$ and the likelihood of the norm statistic.

The results are reported in Table 2. We find that Kolmogorov-Smirnov and Cramér–von Mises both succeed in reporting a lower 99.9th percentile p-value for each tested failure case than the $0.1$th percentile assigned to real samples. Kolmogorov-Smirnov did so with the largest gap, reporting extremely low p-values for all failure cases while reporting calibrated p-values for the positive case. Shapiro-Wilk, which in the literature often is regarded as one of the most powerful tests (Razali et al., 2011), did not work well in our context in contrast to the other normality testing methods, as it produced high p-values also for several of the failure cases. An explanation may be that in our context we have sample sizes of tens of thousands (same as the dimensionality of the latent $D$), while comparisons in the literature typically focuses on smaller sample sizes, such as up to hundreds or a few thousand (Razali et al., 2011). The Chi-square test is a test for discrete distributions. However, it is commonly used on binned data (Larsen & Marx, 2005) to approximate a continuous distribution as discrete. We do this, using histograms of 30 bins. This test successfully produced very low p-values for each failure case, but also did so for many valid samples, yielding a $0.1\%$ of 0. The likelihood $\mathcal{N}(\boldsymbol{x}_*; \boldsymbol{\mu}, \boldsymbol{\Sigma})$ assigns high likelihoods to vectors $\boldsymbol{x}_*$ near the mode irrespective of its characteristics. We note that it fails to distinguish to failure cases from real samples, and in some cases assigns much higher likelihood to the failure cases than real data. The insufficiency of the likelihood to describe the feasibility of samples is a phenomenon present in cases of high-dimensionality and low joint dependence between these dimensions; this is due to the quickly shrinking concentration of points near the centre (or close to $\boldsymbol{\mu}$, where the likelihood is highest) as the dimensionality is increased (Talagrand, 1995; Ash, 2012; Nalisnick et al., 2019). The norm statistic we note successfully assigns low likelihoods to some failure cases, where the norm is atypical, but fails to do so in most of the tested cases.

In summary, we find that Kolmogorov-Smirnov and Cramér–von Mises both correctly rejects the failure cases (by assigning low p-values) while reporting calibrated p-values for real samples, with **Kolmogorov-Smirnov** doing so with the lowest p-values for all failure cases.

## F INTERPOLATION AND CENTROID DETERMINATION SETUP DETAILS

**Baselines** For the application of interpolation, we compare to linear interpolation (LERP), spherical linear interpolation (SLERP), and Norm-Aware Optimization (NAO) (Samuel et al., 2023), a recent approach which considers the norm of the noise vectors. In contrast to the other approaches which only involve closed-form expressions, NAO involves a numerical optimization scheme based on a discretion of a line integral. For the application of centroid determination we compare to NAO, the Euclidean centroid $\bar{\boldsymbol{x}} = \frac{1}{K} \sum_{k=1}^K \boldsymbol{x}_k$, and two transformations of the Euclidean centroid; "standard-ised Euclidean", where $\bar{\boldsymbol{x}}$ is subsequently standardised to have mean zero and unit variance, and "mode norm Euclidean", where $\bar{\boldsymbol{x}}$ is rescaled to have the norm equal to the (square root of the) mode

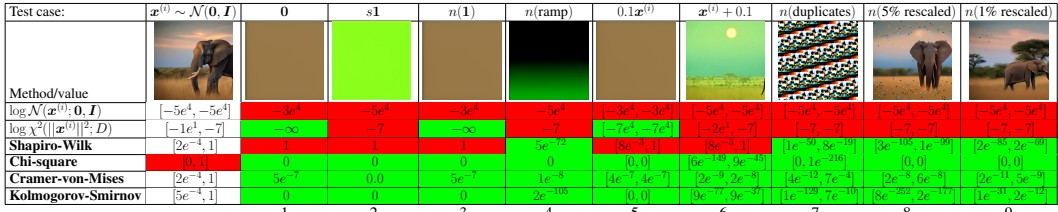

| Test case: | $x^{(i)} \sim \mathcal{N}(0, I)$ | 0 | s1 | n(1) | n(ramp) | $0.1x^{(i)}$ | $x^{(i)} + 0.1$ | n(duplicates) | n(5% rescaled) | n(1% rescaled) |
|---|---|---|---|---|---|---|---|---|---|---|
| **Method/value** | | | | | | | | | | |
| $\log \mathcal{N}(x^{(i)}; 0, I)$ | $[-5e^4, -5e^4]$ | $-3e^4$ | $-5e^4$ | $-3e^4$ | $-5e^4$ | $[-3e^4, -3e^4]$ | $[-5e^4, -5e^4]$ | $[-5e^4, -5e^4]$ | $[-5e^4, -5e^4]$ | $[-5e^4, -5e^4]$ |
| $\log \chi^2(\|x^{(i)}\|^2; D)$ | $[-1e^1, -7]$ | $-\infty$ | $-7$ | $-\infty$ | $-7$ | $[-7e^1, -7e^4]$ | $[-2e^1, -7]$ | $[-7, -7]$ | $[-7, -7]$ | $[-7, -7]$ |
| **Shapiro-Wilk** | $[2e^{-4}, 1]$ | 1 | 1 | 1 | $5e^{-72}$ | $[8e^{-3}, 1]$ | $[5e^{-3}, 1]$ | $[1e^{-50}, 8e^{-19}]$ | $[3e^{-105}, 1e^{-99}]$ | $[2e^{-85}, 2e^{-69}]$ |
| **Chi-square** | $[0, 1]$ | 0 | 0 | 0 | 0 | $[0, 0]$ | $[6e^{-149}, 9e^{-45}]$ | $[0, 1e^{-216}]$ | $[0, 0]$ | $[0, 0]$ |
| **Cramer-von-Mises** | $[2e^{-4}, 1]$ | $5e^{-7}$ | $0.0$ | $5e^{-7}$ | $1e^{-8}$ | $[4e^{-7}, 1e^{-7}]$ | $[2e^{-9}, 2e^{-8}]$ | $[4e^{-12}, 7e^{-4}]$ | $[2e^{-8}, 6e^{-8}]$ | $[2e^{-14}, 5e^{-9}]$ |
| **Kolmogorov-Smirnov** | $[5e^{-4}, 1]$ | 0 | 0 | 0 | $2e^{-105}$ | $[0, 0]$ | $[9e^{-77}, 9e^{-34}]$ | $[1e^{-129}, 7e^{-10}]$ | $[8e^{-252}, 2e^{-177}]$ | $[1e^{-31}, 2e^{-12}]$ |
| | | 1 | 2 | 3 | 4 | 5 | 6 | 7 | 8 | 9 |

Table 2: **Normality testing of latents** Reported is the $[0.1\%, 99.9\%]$-confidence interval of the p-value produced by each respective normality testing method on $1e^4$ random samples in the stochastic test cases. We also report the log likelihood of the vector $x^{(i)}$ under the latent distribution and the likelihood of the norm statistic. The diffusion model in Rombach et al. (2022) is used in this example. Green colour for the respective failure test case indicate the method assigning a lower p-value or likelihood to the failure case to at least $99.9\%$ of the failure samples than the lowest $0.1\%$ of real samples, to illustrate that it was successful in distinguishing the failure samples. Chi-square assigns low p-values to many real samples as well, which we indicate in red. The images show the generated image from a random failure sample of the test case. In (failure) Test 1 in column 1 the latent is $0$, the mode of the latent distribution for this model $\mathcal{N}(0, I)$. In Test 2 it is the constant vector with the maximum likelihood norm of the norm sampling distribution. In Test 3 the constant vector is instead normalised to have its first two moments matching the latent's marginal distribution. $n(\cdot)$ indicate that the vector(s) of the test case is normalised to have zero mean and unit variance. In Test 4 a linspace vector $[-1, \ldots, 1]^D$ have been similarly normalised. In Test 5 and 6 the normal samples (from the left most column) have been transformed with scaling and a bias, respectively. In Test 7 random selections of $1\%$ of the dimensions $[1, \cdots, D]$ of the normal samples have been repeated (100 times) and subsequently normalised by $n(\cdot)$. In Test 8 and 9 a proportion of the dimensions of the normal samples have been multiplied by $5$ (with indices selected randomly) and the whole vectors are subsequently normalised by $n(\cdot)$, where the proportion is $5\%$ and $1\%$, respectively.

of $\chi^2(D)$, the chi-squared distribution with $D$ degrees of freedom, which is the maximum likelihood norm given that $x$ has been generated from a unit Gaussian with $D$ dimensions.

**Evaluation sets** We closely follow the evaluation protocol in Samuel et al. (2023), where we base the experiments on Stable Diffusion 2 (Rombach et al., 2022) and inversions of random images from ImageNet1k (Deng et al., 2009). We (uniformly) randomly select 50 classes, each from which we randomly select 50 unique images, and find their corresponding latents through DDIM inversion (Song et al., 2020a) using the class name as prompt. We note that the DDIM inversion can be sensitive to the number of steps. Therefore, we made sure to use a sufficient number of steps for the inversion (we used 400 steps), which we then matched for the generation; see Figure 15 for an illustration of the importance of using a sufficient number of steps. We used the guidance scale 1.0 (i.e. no guidance) for the inversion, which was then matched during generation. Note that using no guidance is important both for accurate inversion as well as to not introduce a factor (the dynamics of prompt guidance) which would be specific to the Stable Diffusion 2.1 model.

For the interpolation setup we randomly (without replacement) pair the 50 images per class into 25 pairs, forming 1250 image pairs in total. In between the ends of each respective pair, each method then is to produce three interpolation points (and images). For the NAO method, which needs additional interpolation points to approximate the line integral, we used 11 interpolation points and selected three points from these at uniform (index) distance, similar to in Samuel et al. (2023).

For the centroid determination setup we for each class form 10 3-groups, 10 5-groups, 4 10-groups and 1 25-group, sampled without replacement per group[1]; i.e. each method is to form 25 centroids per class total, for an overall total of 1250 centroids per method. Similarly to the interpolation setup, for NAO we used 11 interpolations points per path, which for their centroid determination method entails $K$ paths per centroid.

---

[1]But with replacement across groups, i.e. the groups are sampled independently from the same collection of images.

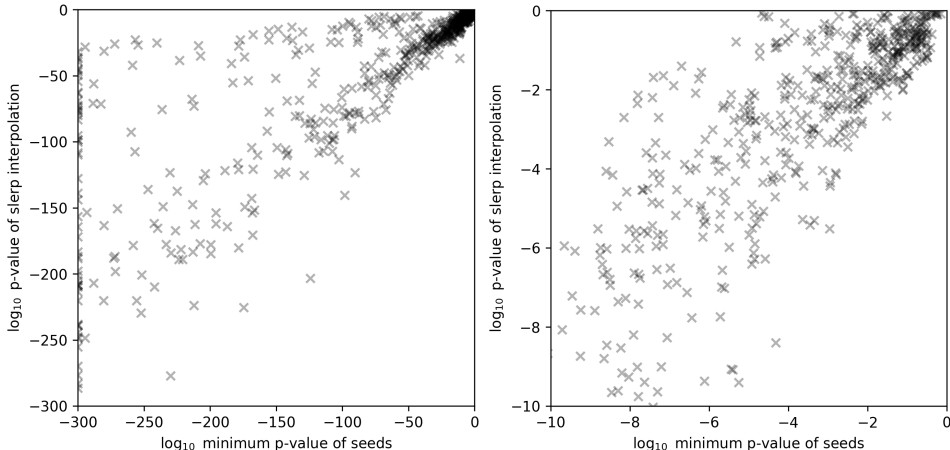

Figure 8: **Low p-values are inherited by interpolants** The x-axis show the lower Kolmogorov-Smirnov p-value *of the two seed latents* for each corresponding spherical interpolation in Figure 5, and the y-axis show the p-value of the *resulting interpolant*. Due to the vast dynamic range of the p-values, the scatter plot is shown in full on the left, and zoomed in on the upper-right quadrant on the right. We note that when any of the two seed latents have a small p-value this largely tend to be inherited by the interpolant. The Pearson correlation coefficient is $0.79$ for the indicator of acceptance (defined as $1$ if the p-value is greater than $1e^{-3}$ and $0$ otherwise) and $0.66$ for the $\log_{10}$ p-values.

**Evaluation**    We, as in Samuel et al. (2023), assess the methods quantitatively based on visual quality and preservation of semantics using FID distances and class prediction accuracy, respectively.

The FID distances are computed using the pytorch-fid library (Seitzer, 2020), using all evaluation images produced per method for the interpolation and centroid determination respectively, to maximize the FID distance estimation accuracy.

For the classification accuracy, we used a pre-trained classifier, the MaxViT image classification model (Tu et al., 2022) as in Samuel et al. (2023), which achieves a top-1 of 88.53% and 98.64% top-5 accuracy on the test-set of ImageNet.

See results in Section H.

## G  ADDITIONAL QUALITATIVE RESULTS

See Figure 9 for a demonstration of subspaces of latents *without* the LOL transformation introduced in Equation 3, using a diffusion model (Rombach et al., 2022) and a flow matching model (Esser et al., 2024), respectively. The setup is identical (with the same original latents) as in Figure 12 and Figure 1, respectively, except without applying the proposed (LOL) transformation.

See Figure 10 for additional slices of the sports car subspace shown in Figure 1.

See Figure 12 and Figure 13 for Stable Diffusion 2.1 (SD2.1) versions of the Stable Diffusion 3 (SD3) examples in Figure 1 and Figure 6, with an otherwise identical setup including the prompts. Just like in the SD3 examples LOL defines working subspaces. However, as expected since SD2.1 is an older model than SD3, the visual quality of the generations is better using SD3.

See Figure 11 for an interpolation example.

In the examples above the SD2.1 and SD3 models are provided with a text prompt during the generation. See Figure 14 for an example where the original latents ($\{\boldsymbol{x}_i\}$) were obtained using DDIM inversion (from images) without a prompt (and guidance scale 1.0, i.e. no guidance), allowing generation without a prompt. This allows us to also interpolate without conditioning on a prompt. We note that, as expected without a prompt, the intermediates are not necessarily related to the end points (the original images) but still realistic images are obtained as expected (except for using linear interpolation, see discussion in Section 4) and the interpolations yield smooth gradual changes.

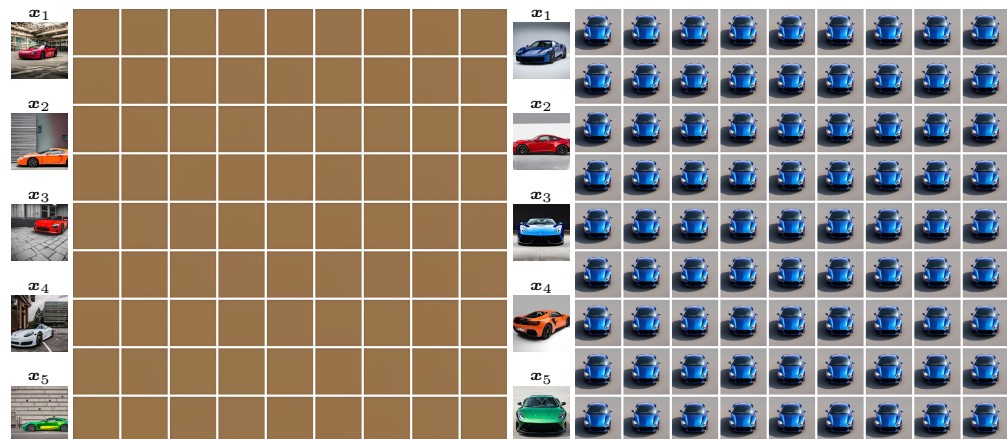

Figure 9: **Without LOL transformation**. The setup here is exactly the same as in Figure 12 and Figure 1, respectively, except without the proposed (LOL) transformation (see Equation 3). The prompt used is *"A high-quality photo of a parked, colored sports car taken with a DLSR camera with a 45.7MP sensor. The entire sports car is visible in the centre of the image. The background is simple and uncluttered to keep the focus on the sports car, with natural lighting enhancing its features."*. We note that the diffusion model does not produce images of a car without the transformation, and neither model produce anything else than visually the same image for all coordinates.

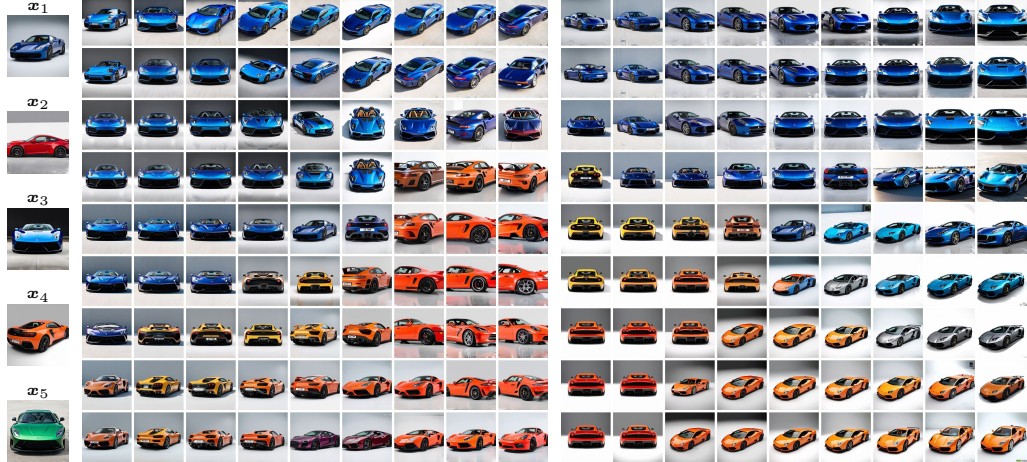

Figure 10: **Additional slices of a sports car subspace**. The latents $x_1, \ldots, x_5$ (corresponding to images) are converted into basis vectors and used to define a 5-dimensional subspace. The grids show generations from uniform grid points in the subspace coordinate system, where the left and right grids are for the dimensions $\{1, 2\}$ and $\{3, 4\}$, respectively, centered around the coordinate for $x_1$. Each coordinate in the subspace correspond to a linear combination of the basis vectors. The flow matching model Stable Diffusion 3 (Esser et al., 2024) is used in this example. See Figure 1 for another slice of the latent subspace.

## H ADDITIONAL QUANTITATIVE RESULTS AND ANALYSIS

In Table 1 we show that our method outperforms or maintains the performance of the baselines in terms of FID distances and accuracy, as calculated using a pre-trained classifier following the evaluation methodology of Samuel et al. (2023). For an illustration of centroids and interpolations, see Figure 2 and Figure 16, respectively. The evaluation time of an interpolation path and centroid, shown with one digit of precision, illustrate that the closed-form expressions are significantly faster than NAO. Surprisingly, NAO did not perform as well as spherical interpolation and several other baselines, despite using their implementation, which was outperforming these methods in Samuel

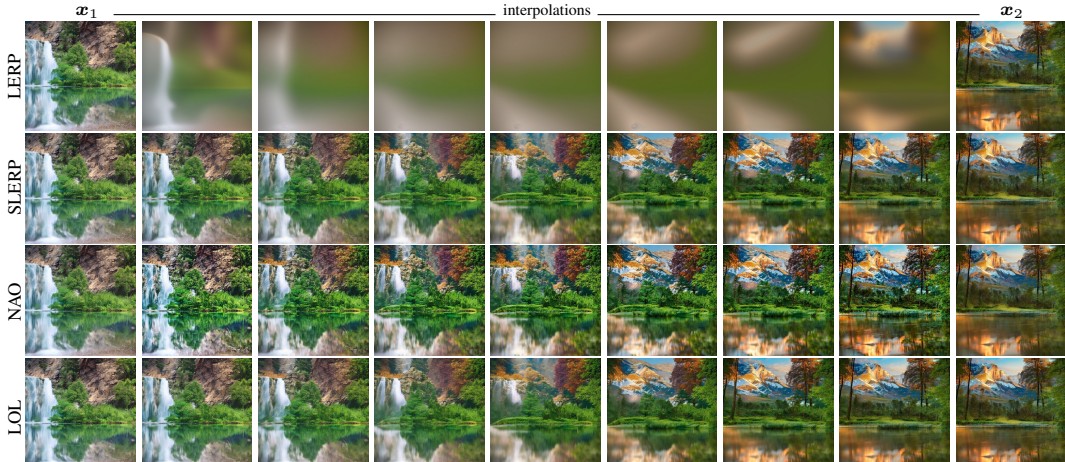

Figure 11: **Interpolation**. Shown are generations from equidistant interpolations of latents $\boldsymbol{x}_1$ and $\boldsymbol{x}_2$ (in $v \in [0, 1]$) using the respective method. The diffusion model Stable Diffusion 2.1 (Rombach et al., 2022) is used in this example.

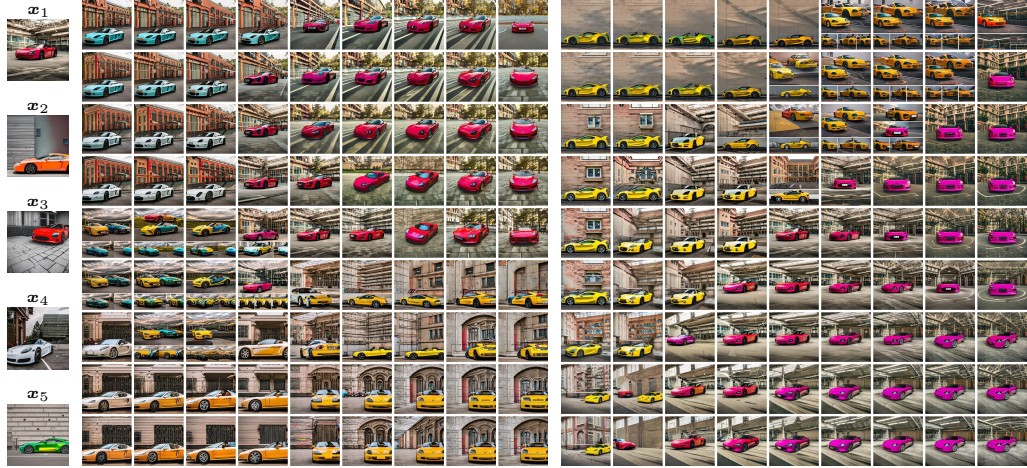

Figure 12: **Low-dimensional subspaces**. The latents $\boldsymbol{x}_1, \ldots, \boldsymbol{x}_5$ (corresponding to images) are converted into basis vectors and used to define a 5-dimensional subspace. The grids show generations from uniform grid points in the subspace coordinate system, where the left and right grids are for the dimensions $\{1, 3\}$ and $\{2, 4\}$, respectively, centered around the coordinate for $\boldsymbol{x}_1$. Each coordinate in the subspace correspond to a linear combination of the basis vectors. The diffusion model Stable Diffusion 2.1 (Rombach et al., 2022) is used in this example.

et al. (2023). We note one discrepancy is that we report FID distances using (2048) high-level features, while in their work they are using (64) low-level features, which in Seitzer (2020) is recommended against as it does not necessarily correlate with visual quality. In Table 3 we report results with all feature settings. We note that, in our setup, the baselines perform substantially better than reported in Samuel et al. (2023) — including NAO in terms of FID distances using the 64 low-level feature setting (1.30 in our setup vs 6.78 in their setup), and class accuracy during interpolation (62% vs 52%), which we study below.

The number of DDIM inversion steps used for the ImageNet images is not reported in the NAO setup (Samuel et al., 2023), but if we use *1 step* instead of 400 (400 is what we use in all our experiments) the class preserving accuracy of NAO for centroid determination, shown in Table 4, resemble theirs more, yielding 58%. As we illustrate in Figure 5 and Figure 15, a sufficient number of inversion steps is critical to acquire latents which not only reconstructs the original image but which can be successfully interpolated. We hypothesise that a much lower setting of DDIM inversions

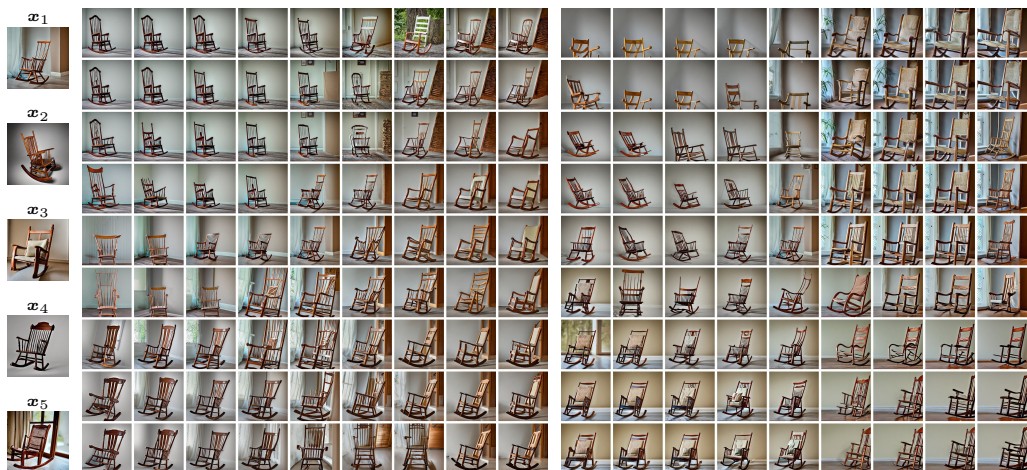

Figure 13: **Low-dimensional subspaces**. The latents $x_1, \ldots, x_5$ (corresponding to images) are converted into basis vectors and used to define a 5-dimensional subspace. The grids show generations from uniform grid points in the subspace coordinate system, where the left and right grids are for the dimensions $\{1, 3\}$ and $\{2, 4\}$, respectively, centered around the coordinate for $x_1$. Each coordinate in the subspace correspond to a linear combination of the basis vectors. The diffusion model Stable Diffusion 2.1 (Rombach et al., 2022) is used in this example.

steps were used in their setup than in ours, which would be consistent with their norm-correcting optimization method having a large effect, making otherwise blurry/superimposed images intelligible for the classification model.

| Interpolation | | | | | | |
|---|---|---|---|---|---|---|
| **Method** | **Accuracy** | **FID 64** | **FID 192** | **FID 768** | **FID 2048** | **Time** |
| LERP | 3.92% | 63.4 | 278 | 2.18 | 199 | $6e^{-3}$s |
| SLERP | 64.6% | 2.28 | 4.95 | 0.173 | 42.6 | $9e^{-3}$s |
| NAO | 62.1% | 1.30 | 4.11 | 0.195 | 46.0 | 30s |
| LOL (ours) | 67.4% | 1.72 | 3.62 | 0.156 | 38.9 | $6e^{-3}$s |
| Centroid determination | | | | | | |
| **Method** | **Accuracy** | **FID 64** | **FID 192** | **FID 768** | **FID 2048** | **Time** |
| Euclidean | 0.286% | 67.7 | 317 | 3.68 | 310 | $4e^{-4}$s |
| Standardized Euclidean | 44.6% | 5.92 | 21.0 | 0.423 | 88.8 | $1e^{-3}$s |
| Mode norm Euclidean | 44.6% | 6.91 | 21.6 | 0.455 | 88.4 | $1e^{-3}$s |
| NAO | 44.0% | 4.16 | 15.6 | 0.466 | 93.0 | 90s |
| LOL (ours) | 46.3% | 9.38 | 25.5 | 0.455 | 87.7 | $6e^{-4}$s |

Table 3: Full table with class accuracy and FID distances using each setting in Seitzer (2020). Note that FID 64, 192, and 768 is not recommended by Seitzer (2020), as it does not necessarily correlate with visual quality. FID 2048 is based on the final features of InceptionV3 (as in Heusel et al. (2017)), while FID 64, 192 and 768 are based on earlier layers; FID 64 being the first layer.

# I  ADDITIONAL IMAGENET INTERPOLATIONS AND CENTROIDS

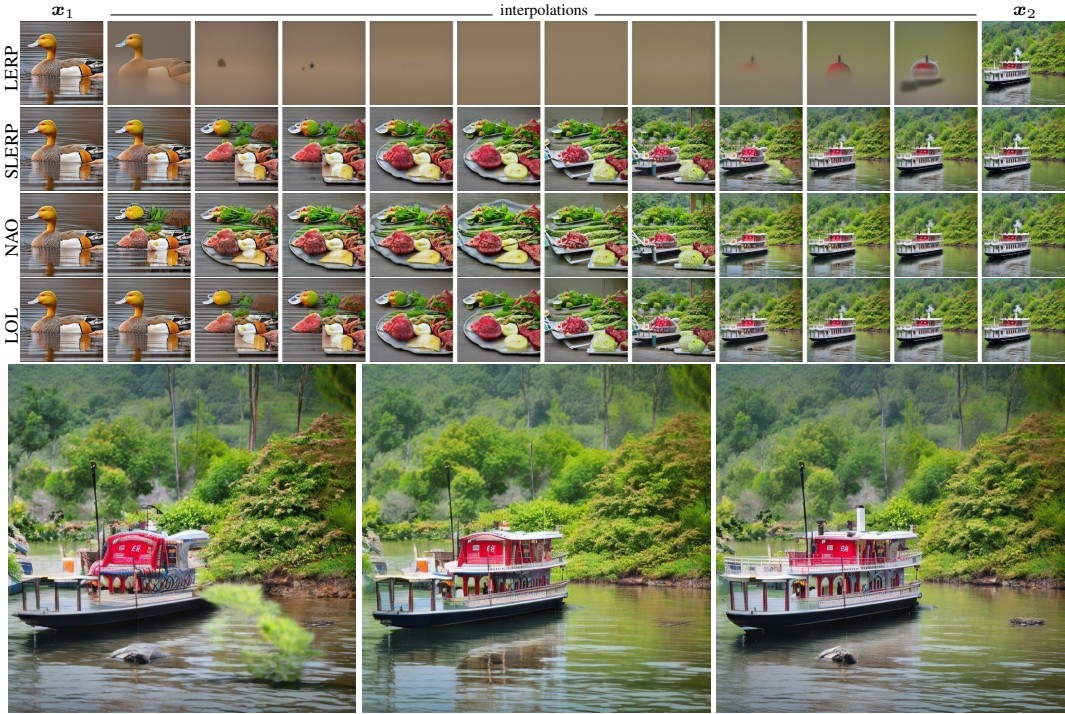

Figure 14: **Interpolation during unconditional generation**. Shown are generations from equidistant interpolations of latents $x_1$ and $x_2$ using the respective method. The latents $x_1$ and $x_2$ were obtained from DDIM inversion (Song et al., 2020a) with an empty prompt, and all generations in this example are then carried out with an empty prompt. The larger images show interpolation index 8 for SLERP, NAO and LOL, respectively. The diffusion model Stable Diffusion 2.1 (Rombach et al., 2022) is used in this example.

| Centroid determination of invalid latents | | | | | | |
|---|---|---|---|---|---|---|
| **Method** | **Accuracy** | **FID 64** | **FID 192** | **FID 768** | **FID 2048** | **Time** |
| Euclidean | 42.3% | 25.7 | 59.3 | 1.23 | 170 | $4e^{-4}$s |
| Standardized Euclidean | 46.6% | 3.35 | 16.6 | 1.23 | 175 | $1e^{-3}$s |
| Mode norm Euclidean | 50.6% | 2.08 | 8.74 | 1.19 | 173 | $1e^{-3}$s |
| NAO | 57.7% | 3.77 | 13.3 | 1.05 | 150 | 90s |
| LOL (ours) | 48.6% | 2.96 | 12.5 | 1.22 | 171 | $6e^{-4}$s |

Table 4: Centroid determination results if we would use a *single step* for both the DDIM inversion and generation (not recommended). This results in latents which do *not* follow the correct distribution $\mathcal{N}(\boldsymbol{\mu}, \boldsymbol{\Sigma})$ and which, despite reconstructing the original image exactly if generating also using a single step, cannot be used successfully for interpolation, see Figure 5. As illustrated in the figure, interpolations of latents obtained through a single step of DDIM inversion merely results in the two images being superimposed. Note that the class accuracy is much higher for all methods on such superimposed images, compared to using DDIM inversion settings which allow for realistic interpolations (see Table 3). Moreover, the FID distances set to use fewer features are *lower* for these superimposed images, while with recommended setting for FID distances (with 2048 features) they are higher (i.e. worse), as one would expect by visual inspection. The NAO method, initialized by the latent yielding the superimposed image, then optimises the latent yielding a norm close to norms of typical latents, which as indicated in the table leads to better "class preservation accuracy". However, the images produced with this setting are clearly unusable as they do not look like real images.

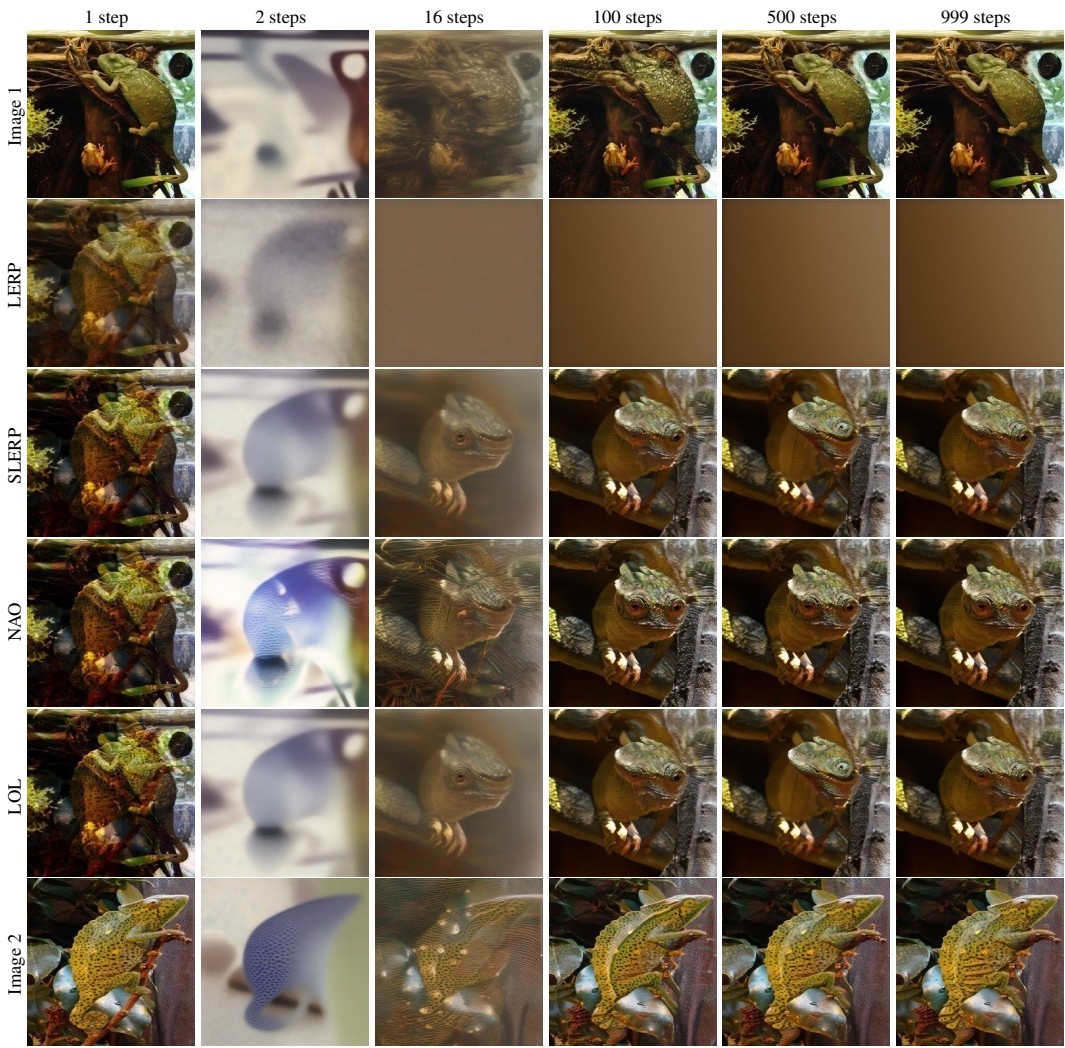

Figure 15: **Accurate DDIM inversions are critical for interpolation**. Shown is the interpolation (center) of Image 1 and Image 2 using each respective method, after a varying number of budgets of DDIM inversion steps (Song et al., 2020a). For each budget setting, the inversion was run from the beginning. We note that although a single step of DDIM inversion yields latents which perfectly reconstructs the original image such latents do not lead to realistic images, but merely visual "superpositions" of the images being interpolated.

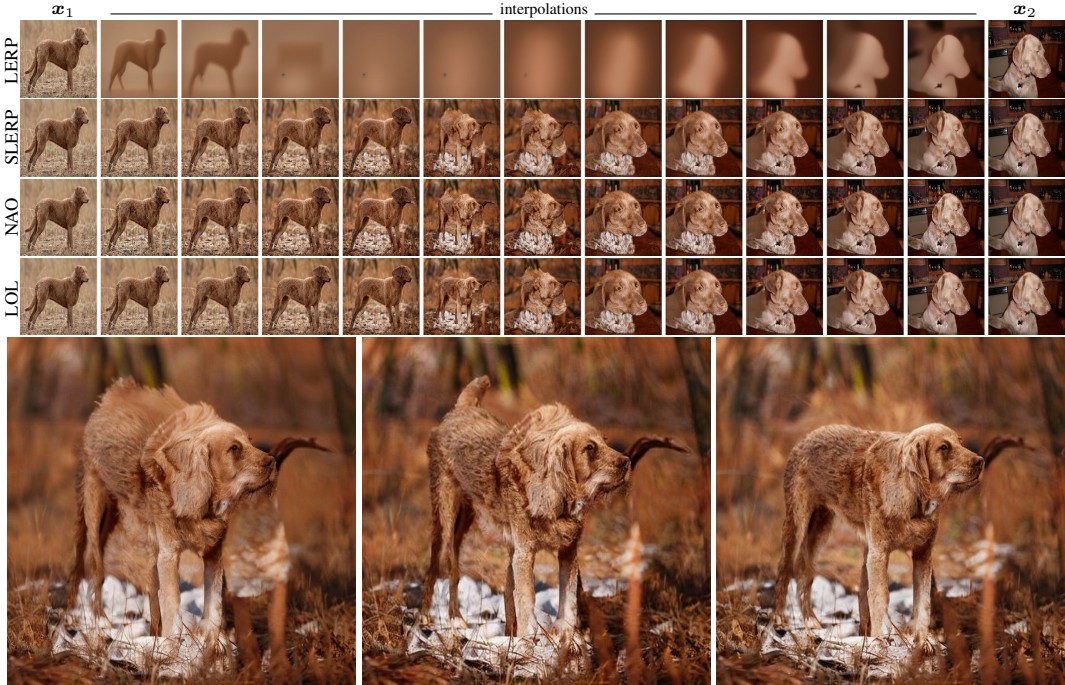

Figure 16: **Interpolation**. Shown are generations from equidistant interpolations of latents $x_1$ and $x_2$ using the respective method. The latents $x_1$ and $x_2$ were obtained from DDIM inversion (Song et al., 2020a) of two random image examples from one of the randomly selected ImageNet classes ("Chesapeake Bay retriever") described in Section 5. The larger images show interpolation index 6 for SLERP, NAO and LOL, respectively. The diffusion model Stable Diffusion 2.1 (Rombach et al., 2022) is used in this example.

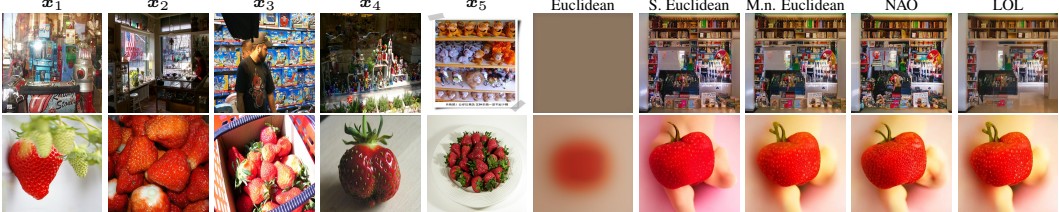

Figure 17: **ImageNet centroid determination**. The centroid of the latents $x_1$, $x_2$, $x_3$ as determined using the different methods, with the result shown in the respective (right-most) plot. The diffusion model Stable Diffusion 2.1 (Rombach et al., 2022) is used in this example.

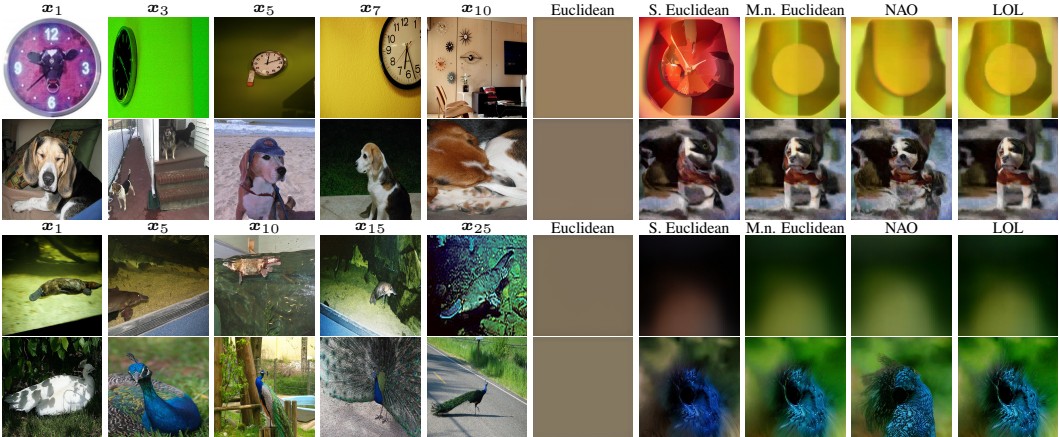

Figure 18: **Centroid determination of many latents**. The centroid of the latents as determined using the different methods, with the result shown in the respective (right-most) plot. Ten and 25 latents are used in the first two and second two examples (rows), respectively, and the plots show a subset of these in the left-most plots. We noted during centroid determination of many latents (such as 10 and 25) that the centroids were often unsatisfactory using all methods; blurry, distorted etc. For applications needing to find meaningful centroids of a large number of latents, future work is needed. The diffusion model Stable Diffusion 2.1 (Rombach et al., 2022) is used in this example.

