# OpenReview forum: "Linear combinations of latents in generative models: subspaces and beyond"
_ICLR.cc/2025/Conference — ICLR 2025 Poster_

### Official Review · Reviewer_ghwF · 2024-11-01

**Soundness:** 4
**Presentation:** 4
**Contribution:** 3
**Rating:** 8
**Confidence:** 4

**Summary:**

This work examines the topic of finding low dimensional subspaces and semantically meaningful interpolant paths between the latent vectors of generative models, such as diffusion and flow models. The authors identify and validate normality tests that confirm their hypothesis that poor visual quality is a product of latent vectors being out of distribution for a model. They propose a simple and easy-to-implement approach for producing interpolants that lead to high quality generation.

**Strengths:**

The paper is very well written. The problem statement, background, identification of the issue, and proposed solution flow well and are easy to follow.

The visual evidence supporting the author’s proposed tests for identifying “failing” latents and for validating their proposed COG approach are compelling. Overall, the experimental validation is thorough and well executed.

**Weaknesses:**

The list below roughly follows the order of appearance in the manuscript. In my opinion W1, W2, W4, and W5 are the most important among these.

### W1: The functional form / actual mechanics of the normality tests should be added as an appendix section
For readers less familiar with the actual tests employed, this would be very helpful. For example, I was only aware of the Kolmogorov-Smirnov test of univariate distributions and am not sure how the multivariate version is defined. I would definitely recommend adding these details as an appendix section to aid clarity of exposition.

### W2: Figure 3 is hard to interpret
I found this figure hard to digest. Specifically, are the images from columns 2 and 3 generated from the same latent vector but with different seeds?

### W3: “Correlation” between acceptance of an inversion and reconstruction quality should be quantified.
In the second paragraph of Section 3.2, the authors state:
> Figure 4 demonstrate a strong positive correlation between acceptance of an inversion (green)
$\mathbf{x}^{(*)}$ and its reconstruction quality,

This claim should be quantified if possible.

### W4: Figure 5 is also difficult to interpret
I found the bottom of Figure 5 also difficult to interpret. The relationship between Q-align score and acceptance under the normality tests does not appear to be clear cut and the confidence intervals seem quite wide.

### W5: The evidence in Section 3.2 seems anecdotal
Specifically the claims made in the third paragraph of this section seem to only be supported by the few examples in the grid of Figure 5. If there were some way to better quantify / make the evidence for these claims more robust, that would make the claim about the efficacy of the normality tests much stronger in my opinion.

### W6: Minor formatting issues
- Line 153: The “Shoemake (1985)” citation should be `\citep`
- Line 158: The “Song (2020a;b)” citations should be `\citep`
- Line 245: The citation for Kolmogorov-Smirnov seems to be malformed: “An 1933” and should also be `\citep`

### W7: Formatting suggestions
List below is not weaknesses per se, just some ideas I had about helping to make some of the figures easier to digest. Feel free to implement or ignore these:
- In Figure 2, and other similar ones (e.g. Figure 6), perhaps place a red / some-colored border around the image in the first column that serves as the “anchor”/origin image and place a similar box around the corresponding image in the grid. Consider labeling the directions in the grid as well.
- In the paragraph for section 3.1, at the end of the paragraph, I would recommend bolding the normality tests that the authors found to be most effective.
- In Figure 5, it might be useful to also add the Q-align score as colored text to each image

**Questions:**

### Q1: In Figure 4, why is inversion with a higher budget worse than 1 step?
Columns with budget 2-32 appear to be worse than a single step. Why is this the case?

---

> ### Author Response · Authors · 2024-11-23
>
> We are delighted to hear that you find the paper very well written and that the experimental validation is thorough and well executed. We hope to address all of your questions below, and look forward to further engaging with you during the remaining rebuttal period if you have any additional questions.
>
> - **W3, W4, W5**: "Figure 5 is difficult to interpret", "If there were some way to better quantify / make the evidence for these claims more robust, that would make the claim about the efficacy of the normality tests much stronger in my opinion."
>
>   Thank you for this feedback. We agreed that Section 3.2 and Figure 5 could be improved to better illustrate the points we are making and we have now addressed this. We have revamped Figure 4 and Figure 5 and the text referencing them to clearly demonstrate the evidence of our claims about the efficacy of normality testing. Please see the updated Section 3.2 along with the updates to the figures.
>
> - **W1**: "The functional form / actual mechanics of the normality tests should be added as an appendix section. For readers less familiar with the actual tests employed, this would be very helpful."
>
>   Thank you for this input. We will add additional background on normality testing to the appendix for the camera-ready version. We have also made improvements to Section 3 to make it clear for readers how to interpret the tests in this setting.
>
> - **W2**: "Figure 3 is hard to interpret. I found this figure hard to digest. Specifically, are the images from columns 2 and 3 generated from the same latent vector but with different seeds?"
>
>   The image in column 2 is generated from the same latent as in column 1 (a "real", explicit sample from the latent distribution) but which has been rescaled to have the most likely norm (under the norm sampling distribution). The latent used in column 3, on the other hand, is merely a scaled constant vector which — despite having the most likely norm *as well as* a very high likelihood under the target distribution itself — does not yield a realistic image.
>
>   This example is to illustrate the point that there are other characteristics beyond its norm that make a vector a "realistic" Gaussian sample, and that these additional characteristics matter to the trained neural network underlying the generative process (whatever these missing characteristics are for the network in question).
>
> - **W6**: "Minor formatting issues"
>
>   Thank you for listing these. They have now all been fixed.
>
> - **W7**: "Formatting suggestions. Not weaknesses per se, just some ideas I had about helping to make some of the figures easier to digest."
>
>   These are great suggestions, thank you. We will include these improvements in the camera-ready version.
>
> - **Q1**: "In Figure 4, why is inversion with a higher budget worse than 1 step?"
>
>   The latent obtained from a single-step inversion looks very similar to the original image itself; it therefore becomes easier to invert it back since the image is already imprinted, resulting in lower reconstruction errors than intermediate budgets. However, such a degenerate latent is highly non-Gaussian and is not useful (e.g., for interpolation) since it cannot meaningfully be combined with other latents, defeating the purpose of the inversion.

---

> > ### Comment · Reviewer_ghwF · 2024-11-24
> > **Response to rebuttal**
> >
> > Thank you for the detailed response. There are still 2 points that I find unclear:
> >
> > **Functional form of tests**
> >
> > You've promised to provide the functional form of the tests in updated manuscripts, but I would appreciate if you include them in this response as well so that I can better understand the method. This is critical to your approach and it feels very odd in my opinion that it is not explicitly spelled out.
> >
> > **Figure 5 is still unclear**
> >
> > I think the new Figure 4 is much clearer now. However the top left/middle of Figure 5 is still somewhat confusing. Is the takeaway supposed to be that there is a really high density of points in the bottom right corner (i.e. low recon error, high p-value)? Perhaps a density plot would be better? Also why did you split this into 2 plots?

---

> ### Author Response · Authors · 2024-11-25
>
> Thank you for asking for clarification.
>
> **Functional form of tests**
>
> Our mistake sorry, we missed a part of your question.
>
> Please see our main comments in the top, specifically "Details on normality testing of latents".
>
> We are able to deploy the standard univariate KS test to assess the latent.
> Specifically, we are using the fact that it is equivalent to test the hypothesis of $\mathbf{x} \sim \mathcal{N}(\mathbf{\mu}, \mathbf{\Sigma}), \mathbf{x} \in \mathbb{R}^D$ for a \textbf{single} observation $\mathbf{x}$ (the latent) as it is to test the hypothesis that a collection of size $D$ of univariate observations $\{\epsilon_d\}_{d=1}^D$ follow $\mathcal{N}(0, 1)$, if $\mathbf{\epsilon} = \mathbf{\Sigma}^{-1}(\mathbf{x} - \mathbf{\mu})$ and $\mathbf{\epsilon} = [\epsilon_1, \epsilon_2, \dots, \epsilon_D]$, which we discuss in Section 3.1 of the paper.
>
> In summary, the KS test is carried out on the sample set $\{\epsilon_d\}_{d=1}^D$ using the standard, univariate, "one-sample" (which means one collection/dataset in distribution-testing terminology) KS test with a unit normal as reference distribution. We use the following scipy implementation
> https://docs.scipy.org/doc/scipy/reference/generated/scipy.stats.kstest.html
>
> **Figure 5 is still unclear**
>
> The top left/middle plot is to show that there are seeds that have low p-values despite having low reconstruction errors. In addition, it is to illustrate the criteria for each respective group of seeds that the interpolations in the bar-plot on the right side are based on, with the group represented with yellow and green colour, respectively. The top left/middle plot is split in two due to the extreme range of the x-axis; note that the left plot have increments of 50 for each tick, while the middle plot use two for each tick. Without this divide it would be difficult to discern points around the "acceptance" threshold (the black dashed vertical line in the middle plot) which divides the low-reconstruction-error seeds into the yellow or green group, respectively.
>
> Please let us know if you have further questions or if anything is still unclear.

---

> > ### Comment · Reviewer_ghwF · 2024-11-25
> > **Thank you for the clarifications**
> >
> > Thank you for these clarifications. I think my original score of 8 reflects the quality of this work and my conclusion that it should be accepted.

---

> > > ### Author Response · Authors · 2024-11-26
> > >
> > > Excellent! Thank you for the constructive feedback and the valuable suggestions.

---

### Official Review · Reviewer_bMMA · 2024-11-03

**Soundness:** 3
**Presentation:** 2
**Contribution:** 2
**Rating:** 5
**Confidence:** 3

**Summary:**

The paper presents Combination of Gaussian variables. It presents the theory behind the method and multiple experiments showing the effectiveness of the proposed method.

**Strengths:**

The method is simple and easy to understand.

**Weaknesses:**

I think the examples in the paper are not convincing that the proposed method is much better than the baselines. Some numbers indicate this but there is no human evaluation in the results. I think that better examples would have made the paper stronger.

**Questions:**

- In Figure 5, it is hard to tell if an interpolation succeeded or failed. I can't really make any clear decision by looking at the images. It would maybe be better to show interpolation between things which are more intuitive such as human faces or animals.
- Would it be possible to have a larger study of the correlation between the p-value and the Q-align values. For example, one can sample many examples and invert them using different budgets, then interpolate between pairs and check visual quality vs p-value,
- In addition, some human evaluation would be good instead of the Q-Align metric.
- I think that for table 1 it would be good to have a MOS evaluation of the different methods, FID is not a good enough metric.
- In Figure 15, all interpolation methods seem to perform equally well. This makes me think that the proposed method is incrementally better at most. This is also the case in table 3.
- What would happen if figure 6 for a different interpolation method? Would it fail?
- There is no limitation section in the paper.
- In figure 14 it seems the CoG generates an image which is remarkably close to image 2 rather than an interpolation of the two images. Could it be that the generated images are just one of the two options image1 and image2? It seems the interpolation is not really an interpolation at all.
- Generally, I think most examples in the paper are not convincing and that examples that demonstrate the method better should have been chosen. For example, an interpolation between a dog and a cat is similar posture, or between two faces, or the same face with a different facial expression.

---

> ### Author Response · Authors · 2024-11-25
>
> We appreciate your thoughtful comments and are pleased to hear that you found our work easy to understand and our method simple. We hope that all your questions are addressed by the answers below, and we look forward to continuing the discussion during the remainder of the rebuttal period should you have additional questions.
>
> **"What would happen if figure 6 for a different interpolation method? Would it fail?"**
>
> Yes, it would fail. Please refer to Figure 9 in the appendix, as mentioned in Section 5.2.
>
> Prior to COG, there were no methods to form subspaces like this. As demonstrated in Figure 9, applying standard linear algebra without the COG transformations produces constant or identical images, depending on the model.
>
> To emphasise this, we have updated Figure 2 in the main paper to illustrate what happens when COG is not used.

---

> ### Author Response · Authors · 2024-11-25
>
> **"I think the examples in the paper are not convincing that the proposed method is much better than the baselines."**
>
> **"In Figure 15, all interpolation methods seem to perform equally well. This makes me think that the proposed method is incrementally better at most."**
>
> Please refer to our main comments at the top of the rebuttal, particularly "COG and spherical interpolation" and "Latent subspaces." The primary focus of our work is not to outperform baselines that are limited to special cases, but to introduce a general methodology that unlocks new capabilities, such as latent subspaces, while maintaining at least equivalent performance for the special cases.
>
> In fact, we prove mathematically that spherical interpolation approximates COG well in the specific context of two-seed interpolation, where spherical interpolation is applicable. We have updated the manuscript to clarify this focus, especially in the revised abstract.

---

> ### Author Response · Authors · 2024-11-25
>
> **"In Figure 5, it is hard to tell if an interpolation succeeded or failed."**
>
> Thank you for this helpful feedback. Other reviewers have raised similar concerns. We have thoroughly revised Figure 5, along with Figure 4 and the associated text in Section 3.2, to make the evidence for our claims much clearer.
>
> **"Would it be possible to have a larger study of the correlation between the p-value and the Q-align values? For example, one can sample many examples, invert them using different budgets, then interpolate between pairs and check visual quality vs. p-value."**
>
> This is precisely the focus of the experiments underlying Figures 4 and 5. Please see the updated versions of these figures for further clarity.

---

> ### Author Response · Authors · 2024-11-25
>
> ### **Human evaluation**
>
> **"some human evaluation would be good", "Good to have a MOS evaluation of the different methods."**
>
> Our experiments involve generating and analysing multiple thousands of images for each method. For instance, the two-seed interpolation experiments alone require evaluating 18,750 generated images (see Appendix E for details) in total. Conducting MOS evaluations with a sufficiently large and diverse sample of human evaluators would be extremely time-consuming and/or would come at significant economic expense.
>
> Additionally, as discussed above, the core focus of our work is on clarifying what the implications of the assumptions of the generative model are on latents and their manipulation, as well as demonstrating new capabilities.

---

> ### Author Response · Authors · 2024-11-25
>
> **"In Figure 14 it seems the CoG generates an image which is remarkably close to image 2 rather than an interpolation of the two images. Could it be that the generated images are just one of the two options image1 and image2?"**
>
> Please examine details in the generated image closely, such as the presence of rocks in the water (absent in the original image), the altered lower floor of the boat, and the red rather than white details. Additionally, the chimney in the interpolation image emits no smoke, unlike image 2.
>
> **"Examples that demonstrate the method better should have been chosen. For example, an interpolation between a dog and a cat in similar posture."**
>
> As discussed above and in our main comments, two-seed interpolation is only a small part of our overall focus. Furthermore, we mathematically prove that spherical interpolation approximates the correct distribution well in this specific case.
>
> Nevertheless, we have included several such interpolations in the appendix for illustrative purposes.
>
> **"There is no limitations section in the paper."**
>
> Thank you for pointing this out. We will add a dedicated limitations section to the camera-ready version of the paper.

---

> ### Author Response · Authors · 2024-11-25
>
> We hope this clarifies your questions and look forward to further engaging with you as the rebuttal process continues. Thank you again for your constructive feedback.

---

> ### Author Response · Authors · 2024-12-02
>
> Do you have any additional questions for us?
>
> We would like to emphasize the key focus areas of this work:
>
> 1. **Demystifying Latent Spaces**: Understanding the latent spaces of diffusion and flow-matching models, and highlighting the importance of ensuring that latents passed to the model are either genuine samples from the assumed latent distribution or exhibit characteristics consistent with such samples.
> 2. **Robustness to Manipulations**: Exploring how we can maintain this consistency under a broad class of manipulations,
> 3. **Unlocking New Capabilities**: Enabling novel capabilities that were previously unrecognized in the literature, such as the ability to define meaningful subspaces within the latent space.
>
> We encourage you to review the joint comments, particularly the sections titled *"COG and Spherical Interpolation"* and *"Latent Subspaces"* at the top of the page.

---

### Official Review · Reviewer_CkyB · 2024-11-03

**Soundness:** 2
**Presentation:** 2
**Contribution:** 2
**Rating:** 6
**Confidence:** 4

**Summary:**

Generative models like diffusion/flow models can map data to Gaussian latent spaces, allowing additional manipulations. However, typical interpolation methods, like linear or spherical interpolation, often fail because they do not fully respect Gaussian distributional properties, leading to unrealistic generations. Current alternatives are either computationally costly or specific to certain models. This paper proposes Combination of Gaussian variables (COG), a simple, generalizable method that maintains Gaussian characteristics during interpolation and linear transformations, enabling effective latent manipulation across models and data types.

**Strengths:**

1. The core argument of the paper, "the norm is not a sufficient statistic to look at when determining whether the latent is good for generations", is intuitive and sound. Especially Figure 3 presents a convincing argument.
2. They paper has many illustrations of the different interpolation methods results. In particular, the ones of low-dimensional subspaces are very nice.

**Weaknesses:**

1. The contribution of this paper is limited. Norm being not enough is known, and the proposed interpolation method, under the setting of diffusion/flow model, is just rescaling the samples norm: $\alpha=0$ and we rescale the sample by the expected norm gain $\sqrt{\beta}$.
2. The normality testing part feels out of place and under-explored. I assume it is there to justify that the norm is not enough, and we need broad characteristics. In that case, Figure 3 alone is enough. Then the paper never specifies what broad characteristics we should/could look at or optimize. Lastly but more importantly, the exploration done in Section 3 never went anywhere: as listed in point 1, the proposed method also does not look at the broad characteristics (essentially still just the norm since $y$ to $z$ is affine), and is not connected to the normality testing in any way.
3. The actual advantages of the method are questionable. For example, there are no clear differences shown in Figure 1, compared to the (standardized/mode norm) Euclidean results. Also, I am wondering why COG is faster than Euclidean methods in Table 1?
4. The writing of this paper can be improved. For example: (1) the writing on normality testing part is not clear. How exactly are you doing the normality test, as it requires a set of data points? (2) what setting is Table 1 for, SD or ImageNet Inversion, since there is only one FID per task?

**Questions:**

1. I am confused by Figures 4 and 5. Is the 1-step inversion image correct? I expect an image of worse quality than a 2-step inversion following the trend. On this note, could you explain how you are doing the inversion, like one-step inversion and multi-step generation? Essentially, I am curious as to why a 1-step inversion reconstruction is this good, and the quality dips first and then becomes better as you increase the budget.
2. The claim made in Section 3.2, "stress some latents are rejected that lead to good reconstructions...", needs to be backed up, in addition to the illustration in Figure 5. There should be more larger-scale quantitative evaluations. Even Figure 5 itself is not very convincing: why is the snake interpolation better than the clock one?
2. Some Figures are not very readable. For example, the colored numbers in Figure 5, and the green crosses and red dots on the bottom row of Figures 4 and 5.

---

> ### Author Response · Authors · 2024-11-24
>
> We are happy to hear that you find one of our key argument convincing and sound, as well as find important figures convincing and nice. We hope to address all of your questions and concerns below, and look forward to further engaging with you during the rebuttal period should you have any additional queries.
>
> 1a. **"Norm being not enough is known"**
>
> We agree that it is well-established in the statistics literature that the norm of a random variable is not sufficient to capture all the information needed to assess its normality. Nonetheless, within the generative model literature—such as in work on diffusion models and flow-matching—the focus has been on the norm to both diagnose and improve interpolation methods. Note that the norm was a primary focus in both spherical interpolation and Samuel et al. (2023).
>
> Please see our main comments above. If you know of work we may have missed that focuses on a broader set of characteristics or normality testing within the context of generative models, we would appreciate it if you could share it. Such references would be valuable for our readers.
>
> **Reference:**
> Dvir Samuel, Rami Ben-Ari, Nir Darshan, Haggai Maron, and Gal Chechik. *Norm-guided latent space exploration for text-to-image generation*. Advances in Neural Information Processing Systems, 37, 2023.
>
> ---
>
> 1b. **"The proposed interpolation method, under the setting of diffusion/flow model, is just rescaling the sample's norm: $\alpha = 0$, and the sample is rescaled by the expected norm gain $\sqrt{\beta}$."**
>
> Please see our main comments above, including the discussion in "Latent Subspaces."
>
> Firstly, although the expressions are simple, determining how much to rescale as a function of the seed latents is critical, as well as the motivations for this treatment. We have not observed the proposed interpretation and treatment of latent manipulations in prior works, which typically resort to auxiliary methods for special cases. Our aim is to clarify and utilize important, already-existing methodological assumptions to simplify how models are used in downstream tasks and enable new capabilities.
>
> Secondly, $\alpha$ is not necessarily zero in all models. If it is not, adjustments need to account for that, as per the expression.
>
> Lastly, a key point of our work is that for a whole class of operations—those expressible as linear combinations—this simple correction suffices. *This assumes* the seed latents follow the correct distribution, which is why Section 3 on normality testing is crucial for helping practitioners assess this.

---

> ### Author Response · Authors · 2024-11-24
>
> 2. **"The normality testing part feels out of place and under-explored."**
>
> Please see Section 3 of the paper, the comments above, and the main comments at the top, especially "Starting and staying on the model-supported manifold" and "Details on normality testing of latents.".
>
> 3. **"The actual advantages of the method are questionable. For example, there are no clear differences shown in Figure 1, compared to the (standardized/mode norm) Euclidean results."**
>
> Please see our main comments above, particularly "COG and Spherical Interpolation" and "Latent Subspaces."
>
> There are differences in Table 1 between the images produced using COG and the standardized/mode norm Euclidean methods. For instance, observe the front-right tire of the car, the headlights, and the floor. You may need to zoom in slightly to notice the distinctions.

---

> ### Author Response · Authors · 2024-11-24
>
> 5. **"I am wondering why COG is faster than Euclidean methods in Table 1?"**
>
> Good question. First, to clarify, the point we intend to make with showing the compute times is the significant difference between NAO (requiring optimization) and the remaining methods, which are all based on analytical expressions. In Section 5.1, we write: "The evaluation time of an interpolation path and centroid, shown with one digit of precision, illustrates that the analytical expressions are significantly faster than NAO." NAO took 30 seconds on average, while the analytical expressions are computed in milliseconds. We do not claim COG is meaningfully faster than the other analytical methods.
>
> Regarding why COG was slightly faster in our evaluation:
> COG was marginally faster in practice than the "standardized" and "mode norm normalized" Euclidean methods, but not the unnormalized Euclidean method. The difference arises due to fewer floating-point operations in evaluating the expressions. Specifically, with COG we compute $\alpha$ and $\beta$ for each linear combination (along the interpolation path, for each subspace coordinate, or at the individual centroid in this case) and then broadcast them during computation of the linear combination. In contrast, the standardized method first calculates the Euclidean mean, then computes its moments, followed by normalization. Similarly, for the mode norm, the original norms  must first be calculated followed by rescaling.
>
> ---
>
> 6. **"The writing on the normality testing part is not clear. How exactly are you doing the normality test, as it requires a set of data points?"**
>
> Please see our main comments above, especially "Details on Normality Testing of Latents."
>
> We leverage the equivalence between testing whether
> $\mathbf{x} \sim \mathcal{N}(\mathbf{\mu}, \mathbf{\Sigma})$,
> where $\mathbf{x} \in \mathbb{R}^D$, for a single observation
> $\mathbf{x}$, and testing whether a collection of size $D$ of
> univariate observations $\{\epsilon_d\}_{d=1}^D$ follows
> $\mathcal{N}(0, 1)$. This equivalence holds when
> $\mathbf{\epsilon} = \mathbf{\Sigma}^{-1/2}(\mathbf{x} - \mathbf{\mu})$
> and $\mathbf{\epsilon} = [\epsilon_1, \epsilon_2, \dots, \epsilon_D]$,
> as discussed in Section 3.1 of the paper.
>
> ---
>
> 7. **"What setting is Table 1 for, SD or ImageNet Inversion, since there is only one FID per task?"**
>
> Stable Diffusion (SD 2.1) is a model, and ImageNet is a labeled dataset of images. For further details, please refer to Section 5.1 and Appendix E. Our experiments use images randomly sampled from 50 randomly selected ImageNet classes. We use the official inversion implementation of SD 2.1 to invert these images.

---

> ### Author Response · Authors · 2024-11-24
>
> ### Questions:
>
> Question 2-3. **Figure 4 and 5 are not very readable. Figure 5 does not convincingly demonstrate the claim in Section 3.2.**
>
> Thank you for your feedback on these figures, which aligns with similar comments from reviewers ghwF and bMMA. We have now overhauled these figures to clarify our points. Please see the updated figures in conjunction with Section 3.2.
>
> ---
>
> 1. **"I am confused by Figures 4 and 5. I am curious as to why a 1-step inversion reconstruction is this good, and the quality dips first and then becomes better as you increase the budget."**
>
> Excellent question, thank you.
>
> The reason a 1-step inversion can yield lower reconstruction errors than mid-budget settings is that the latent obtained in the former case is very similar to the original image itself. This makes it easier to invert back since the image is already imprinted. However, such a latent is highly non-Gaussian and is not useful (e.g., for interpolation) as it cannot meaningfully be combined with other latents, defeating the purpose of the inversion.
>
> That said, we originally matched the number of generation steps to the number of inversion steps in this experiment for each setup, as this empirically—at least within the context of Stable Diffusion 2.1—yields slightly better reconstructions at all inversion budgets, but particularly at lower budgets. This may be explained by the trajectory induced during inversion (although leading to an invalid, degenerate latent at low budgets) being more closely followed "on its way back" during generation. This effect becomes particularly pronounced in the single-step case, leading to an almost perfectly restored image but, as discussed, a degenerate, image-like latent.
>
> However, this setup made the interpretation of the experiment less straightforward than it could be, as the effect of a lower inversion budget became entangled with the effect of a lower generation budget.
>
> In connection with updating the figures (see the above question), we re-ran the underlying experiments to use the maximum generation budget allowed by the model (999 steps) for every inversion budget to isolate the effect of the latent during the generation process. In practice, the effect on the results was very small with respect to the key claims we are making. Nonetheless, the side-effects you pointed out were unnecessarily distracting for readers, and the updated experiment methodology helps clarify the interpretation.

---

> > ### Comment · Reviewer_CkyB · 2024-12-02
> >
> > Thank you for the response. I acknowledge the contribution of proposing COG from a more distributional perspective, which contrasts prior works that are more empirical observation driven. Thus, I am raising my score to 6.

---

### Official Review · Reviewer_Y6W4 · 2024-11-04

**Soundness:** 2
**Presentation:** 2
**Contribution:** 2
**Rating:** 5
**Confidence:** 3

**Summary:**

This paper proposed a novel method of linearly combining Gaussian latents of pretrained generative models (in the paper Diffusion model and flow matching model) that are simple, and shown in to be effective in linear interpolation and centroid determination tasks. The methodology is based on observation that latent samples from priors of pretrained generative models can lack Gaussian characteristics, which is the main reason for failure in the image reconstruction task.

**Strengths:**

- Well-motivated problem, the proposed method is novel, reasonable, and most importantly, relatively simple, yet seems to performing quite well in (certain) empirical benchmarks.

**Weaknesses:**

- The writing of the paper can be improved. Some parts are ambiguously written. For example, the notion of latent space $x_T$ can be unified for both flow matching and diffusion at time $T$, instead of writing $x(0)$ and $x_T$ to avoid confusion. The important observation, starting at line 206: "Having a latent vector with a norm that is likely for a sample from..." is extremely unclear. In fact, the sentence does not make sense.
- I have a big question mark about the usage of the univariate testing method in Section 3 (stated in 3.1 in between line 250-254). Why don't the authors use a multivariate hypothesis testing method that accounts for the correlation between features/pixels. This makes more sense statistically.
- The authors mentioned that "In this work we will propose a simple technique that guarantees that interpolations follow the latent distribution given that the original (seed) latents do, and which lets us go beyond interpolation". However, I failed to find a Lemma/Theorem that stated such a theoretical guarantee of their method.
- Empirical results seem questionable: the authors claimed to follow closely benchmark protocols from NAO paper of Samuel et al. (2023) (in fact the year of publication for this citation in the paper was stated wrongly -- please fix). More specifically, the results from Table 1 of the paper is very different from Table 1 in Samuel et al. (2023) -- the two tables are supposed to report results from the exact same experiment.

Dvir Samuel, Rami Ben-Ari, Nir Darshan, Haggai Maron, and Gal Chechik. Norm-guided latent space exploration for text-to-image generation. Advances in Neural Information Processing Systems, 37, 2023.

**Questions:**

- Can COG perform rare-concept generation experiment similar to Samuel et al. (2023), section 5.1? I find the current empirical benchmarks that make comparisons with existing baselines quite lackluster.

---

> ### Author Response · Authors · 2024-11-24
>
> We are happy to hear that you find our work novel, reasonable, and well performing despite its relative simplicity. We hope to address all of your questions and concerns below. We look forward to further engaging with you during the remaining part of the rebuttal period, should you have any additional queries.
>
> - **"The authors mentioned that 'In this work we will propose a simple technique that guarantees that interpolations follow the latent distribution given that the original (seed) latents do, and which lets us go beyond interpolation'. However, I failed to find a Lemma/Theorem that stated such a theoretical guarantee of their method."**
>
> Please see Appendix A, Lemma 1 and Lemma 2 for the proofs. These are referred to in Section 4.
>
> - **"The results from Table 1 of the paper is very different from Table 1 in Samuel et al. (2023) -- the two tables are supposed to report results from the exact same experiment."**
>
> Thank you for pointing out the incorrect year in the citation; we have updated that now.
>
> Indeed, the results are different compared to Samuel et al. (2023). We discuss and ablate the discrepancy between our results and those presented in the NAO paper in Appendix G (referred to in Section 5.1 of the paper), and present multiple additional results as part of this analysis.
>
> For example, we show the results using various FID settings. We note that their method is only competitive when using FID based on (64) low-level features (which is the setting they report in their paper), not FID based on (2048) high-level features as per standard FID. See Seitzer (2020) for the FID implementation used. The setting of low-level features is discouraged in Seitzer (2020) since the scores might "no longer correlate with visual quality". Importantly, we also note that NAO *performs better* in absolute terms in *our paper* for interpolation than in the original paper, its just that the baselines perform *even better*, leading to a change in the ordering of results.
>
> The only experiment and metric in which the NAO method performed better in their setup compared to their method in our setup was for the "class-preserving" accuracy specifically for centroid determination, while the standard FID score (using high-level features) in the same experiment was worse than the other methods.
>
> Our hypothesis is that they used fewer inversion steps to obtain the latents being interpolated than in our experiments, which, as per the discourse in our paper, may lead to low-quality latents. We do not know the number of steps they used for inversion since it's not reported in their paper. To test this hypothesis we ran all results for centroid determination again, but where only a single inversion step is used to obtain the latents, which yields highly non-Gaussian latents which are very similar to the images themselves. The results of this are reported in Table 4 in the appendix. In this setting, the methods order/rank in terms of accuracy is largely flipped, and their method yields the largest "accuracy" in terms of yielding interpolated images predicted to be the same class as the endpoints. However, the images yielded from the interpolation of such degenerate latents is merely a superposition of the original images -- see Figure 14 in the appendix. Such images (as demonstrated from the accuracy numbers) makes it relatively easy for the classifier to predict the class, where the visual features of the original images are still present albeit superimposed. Meanwhile, the superimposed images themselves are clearly not useful, as they correspond to trivial addition of the image pixel values. We further note, as discussed in the caption of Table 4, that the FID score using the non-standard setting of 64 low-level features is *better* for these degenerate, superimposed images than the original images, but worse using FID with the standard (2048) high-level features, as one would expect by visual inspection.
>
> Dvir Samuel, Rami Ben-Ari, Nir Darshan, Haggai Maron, and Gal Chechik. Norm-guided latent space exploration for text-to-image generation. Advances in Neural Information Processing Systems, 37, 2023.
>
> Maximilian Seitzer. pytorch-fid: FID Score for PyTorch. <https://github.com/mseitzer/pytorch-fid>, August 2020. Version 0.3.0.

---

> ### Author Response · Authors · 2024-11-24
>
> - **"The notion of latent space can be unified for both flow matching and diffusion at time, instead of writing $\textbf{x}(0)$ and $\textbf{x}_T$ to avoid confusion."**
>
> Thank you for this feedback. In Section 2.1 we explain the difference in the typical notation used in the diffusion model and flow-matching literature, and do unify them when we set up our notation. However, we take your point that we could just as well unify them directly as they are introduced. We have changed that now.

---

> ### Author Response · Authors · 2024-11-24
>
> - **"Having a latent vector with a norm that is likely for a sample from..." is extremely unclear. In fact, the sentence does not make sense."**
>
> "A likely norm" refers to a norm having a high likelihood under its sampling distribution. For example, for a random variable $\textbf{x} \sim \mathcal{N}(\textbf{0}, \textbf{I}), \textbf{x} \in \mathbb{R}^D$, a classic result is that the squared norm is distributed as the chi-squared distribution with $D$ degrees of freedom, $||\textbf{x}||^2_2 \sim \chi(D)$.
>
> We have now updated the paragraph underneath the statement to clarify what is meant.

---

> ### Author Response · Authors · 2024-11-24
>
> - **"Why don't the authors use a multivariate hypothesis testing method that accounts for the correlation between features/pixels."**
>
> Please see our main response, in particular "Details on normality testing of latents".
>
> We do not use a multivariate hypothesis testing method because in our special case of a single ($D$-dimensional) vector observation we are able to — via standardization (using the inverse of the covariance matrix, which is typically trivially tractable in this setting) — equivalently test it using univariate methods (as a collection of unit normal observations).  As of the equivalence, we do indeed account for correlations between the dimensions.
>
> Importantly, applying a standard multivariate testing method to a dataset/collection (called a "sample" in distribution-testing terminology) of a single high-dimensional ($D$) observation is impractical and typically infeasible, while a (univariate) test testing a collection ("sample") of $D$ observations for unit normality is trivial in contrast - meanwhile nothing is lost due to the equivalence in this special case.

---

> ### Author Response · Authors · 2024-11-24
>
> ### Questions:
>
> - **"Can COG perform rare-concept generation experiment similar to Samuel et al. (2023), section 5.1? I find the current empirical benchmarks that make comparisons with existing baselines quite lackluster."**
>
> We have not investigated COG on rare-concept generation as it is outside the scope of our work.
>
> Note that we do not propose our methodology for improving support for concepts poorly supported by the generative model. If a pre-trained model has poor support for certain data you are interested in, or you wish to have a different "layout" of the data representation in the latent space (e.g., encoded through a new, down-stream metric), this is outside the scope of our work; we merely provide a methodology to index the supported manifold of the latent space as per the generative model (please see our main response). If you need custom behaviours for rare concepts, that is an auxiliary task aligned with the work of Samuel et al. (2023) or Samuel et al. (2024) by the same authors.
>
> Dvir Samuel, Rami Ben-Ari, Nir Darshan, Haggai Maron, and Gal Chechik. Norm-guided latent space exploration for text-to-image generation. *Advances in Neural Information Processing Systems*, 37, 2023.
>
> Samuel, Dvir, et al. "Generating images of rare concepts using pre-trained diffusion models." *Proceedings of the AAAI Conference on Artificial Intelligence*. Vol. 38. No. 5. 2024.

---

> > ### Comment · Reviewer_Y6W4 · 2024-11-27
> >
> > Thank you for your extensive rebuttal and my apologies for being late to acknowledge it. I am raising my score to borderline (5) that reflects my remaining doubts about the empirical evaluation of the paper (only one table of quantitative metrics is provided, the rest are qualitative images). I will further discuss with other reviewers to make a final conclusion.

---

> > > ### Author Response · Authors · 2024-11-29
> > >
> > > Thank you for your feedback.
> > >
> > > We conduct several quantitative experiments, with results presented in tables (Table 2, 3, and 4) and others visualised in plots (Figure 4, 5, and 8).
> > >
> > > Could you kindly let us know if there are specific areas where additional empirical evaluation would strengthen the paper? We greatly appreciate any suggestions on how we might further verify our claims.

---

> > > ### Author Response · Authors · 2024-12-02
> > >
> > > We like to highlight that the focus of the work is:
> > > 1. demystifying latent spaces of diffusion and flow-matching models and stressing the importance of making sure that latents passed to the model always are samples from the assumed latent distribution or have characteristics that match those of such samples,
> > > 2. how we can ensure this holds under a general class of manipulations, in order to
> > > 3. unlock new capabilities previously not known in the literature to be possible, such as defining subspaces of the latent space.
> > >
> > > Please see the joint comments, especially "COG and spherical interpolation" and "Latent subspaces" in the top of the page.
> > >
> > > There are no baselines for new capabilities like latent subspaces to meaningfully compare quantitatively since our work is first to demonstrate such capabilities. We show in Figure 2 (please see the updated paper) and Figure 9 that, without the proposed COG transformation, the subspace coordinates produce identical or single-colour images.

---

### Author Response · Authors · 2024-11-18

We sincerely thank all the reviewers for their thoughtful and detailed feedback. We hope to below have addressed some overarching concerns, clarified misunderstandings, and highlighted the unique contributions of our work. We have also responded to each reviewer’s specific questions separately.

**Key contributions**
To clarify, the focus of our work is threefold:
1. demystifying latent spaces of diffusion and flow-matching models and stressing the importance of making sure that latents passed to the model always are samples from the assumed latent distribution or have characteristics that match those of such samples,
2. how we can ensure this holds under a general class of manipulations, in order to
3. unlock new capabilities previously not known in the literature to be possible, such as defining subspaces of the latent space.

Points (1) and (2) were identified by the reviewers, but (3) we have now emphasised in the paper.

Based on feedback from the reviewers we have updated the abstract, Figure 4 & 5 along with the discussion of these figures in Section 3.2, as well as updated the notation slightly in Section 2. We think these changes greatly improves the clarity and the potential impact of our paper, and are grateful to the reviewers for their feedback.

---

> ### Author Response · Authors · 2024-11-18
> **COG and spherical interpolation**
>
> COG lets us operate with any linear combination of any number of (supported) latents and still yield supported latents, instead of merely being able to interpolate between two (supported) latents as with spherical interpolation. Critically, we are not suggesting that COG works notably better than spherical interpolation (SI) \emph{in the context of two-latent interpolation} for which SI is available. In fact, we prove in Appendix B (which we refer to in Section 4) that spherical interpolation yields latents which are very close to following the correct distribution --- which is the first theoretical motivation, to our knowledge, which holistically shows why it works at all. In the paper where it was originally proposed for generative models White (2016) it was motivated heuristically and empirically, and in Kilcher (2017) it was motivated as a way to the keep norms of  interpolants close to the norms of the endpoints (motivated by the sampling distribution of norms for Gaussian samples), which is only part of the story. Our theoretical result is consistent with SI being known to work well empirically, and with SI performing very similar to COG in our (two-latent) interpolation experiments.
>
> However, as discussed, the motivation and focus of our paper is to go beyond such interpolation, to enable capabilities which were previously not available. Moreover, with our experiment we empirically demonstrate what we had shown theoretically, that we do not need to treat the special case of two-latent interpolation with spherical interpolation, but can deploy our --- much more general, now exact, and still simple to implement --- method also for this special case. We will make sure this is clear in the camera-ready version.
>
> White, Tom. "Sampling generative networks." Advances in Neural Information Processing Systems 29 (NIPS 2016).
>
> Kilcher, Yannic, Aurélien Lucchi, and Thomas Hofmann. "Semantic interpolation in implicit models." arXiv preprint arXiv:1710.11381 (2017).

---

> ### Author Response · Authors · 2024-11-18
> **Latent subspaces**
>
> As discussed above, although COG does match or only trivially exceeds the performance of spherical interpolation, COG is a significantly more general approach to latent manipulation --- allowing the creation of model-supported latents from any linear combination of any number of seed latents. We showcase an important example of such a new capability, which is defining subspaces of the latent space via data examples, applicable to any pre-trained generative model with Gaussian latents and without any need for further training. Note that prior to COG there was no obvious way to build subspaces, as demonstrated in Figure 8 of the appendix, where without the proposed transformation all subspace coordinates merely yield identical and/or block colour images. In contrast, with COG, all subspace coordinates yield valid images expressing a diverse set of features based on the original seeds. Based on your feedback we plan to highlight Figure 8 for readers within the main part of the paper.
>
> The working latent subspaces are achieved by using COG with the corresponding linear combination weights that uniquely identifies each individual coordinate in the subspace; with derivations and proofs in Appendix A for COG and Appendix C for the linear combination weights corresponding to the subspace coordinates. As of the subspace capability afforded via COG, we are now able to index low-dimensional latent subspaces of interest, a capability with no baselines since no such method previously existed. Our work opens up for diffusion and flow-matching models to be used in new settings by combining it with existing methodologies, such as search and optimization (e.g. Bayesian Optimization), since search now can be readily defined directly over targeted subspaces as if they were Euclidean, mapped smoothly via the COG to the model-supported manifold. It is important to note that although COG is an affine transformation with respect to any particular $\mathbf{y} = \sum_{k=1}^K w_k \mathbf{x}_k$ (where $\mathbf{y}$ is the original naive linear combination of latents $\mathbf{x}_k$, following a Gaussian with incorrect mean and/or covariance), COG produces a different such transformation depending on the particular location in the space (determined through a non-linear function), yielding an overall non-affine map to the model-supported latents.

---

> ### Author Response · Authors · 2024-11-18
> **Starting and staying on the model-supported manifold**
>
> As discussed above, the critical consideration we stress is having latents which have characteristics that match those of samples of the model's latent distribution. For a complete treatment, we need to make sure to both (1) \textbf{start} with valid latents, i.e. where the latents we are to manipulate have the characteristics of samples of the correct distribution to begin with, as diagnosed by normality testing of inversions, and (2) \textbf{maintain} this when operating on the latents, as provided by our proposed COG. Spherical interpolation, as we discuss in Section 4 and prove in Appendix B, \emph{already} (approximately, but closely) follows the correct distribution \emph{given that} the starting latents --- the endpoints of the interpolation --- are valid to begin with. In contrast, as discussed in the paper, several prior works (including NAO) have been motivated by the empirical observation that spherical interpolation does not seem to work well on latents obtained from inversion; in our paper we show that there is nothing special about inverted latents except that the inversion may fail to produce valid latents. As we show in experiments, such failures happen frequently even with a widely popular diffusion model (SD 2.1) and its official inversion implementation (also relied on in the NAO paper) --- which can happen due to insufficient settings and to some extent even using the maximum (supported) settings. Normality testing is proposed to detect when the latents resulting from inversion will fail to provide effective interpolation so that it can be discovered and treated at the source.

---

> ### Author Response · Authors · 2024-11-18
> **Details on normality testing of latents**
>
> We will here discuss our Section 3 on normality testing, clarifying some details around the normality testing setup which received questions from reviewers Y6W4, CkyB, and ghwF.
>
> We propose normality testing to detect if latents acquired from inversion fail to have characteristics of samples following the correct distribution $\mathcal{N}(\mathbf{\mu}, \mathbf{\Sigma})$. As discussed in the paper --- due to how the generative model is trained and that the generative model is defined as a  transformation of latent vectors from a particular Gaussian distribution to the data distribution, where a neural network defines this transformation --- model support requires that the latents have the characteristics of such samples that the neural network has come to expect.
>
> The specific characteristics actually expected by the neural network importantly: (1) depend on the specific neural network used, including its architecture, and (2) can be complicated subtleties, far exceeding simple characteristics such as norms, since the neural network has implicitly extracted them during its training.
>
> As discussed in the paper, to avoid the need to design specific tests for each specific trained generative model (and neural network), we propose using tests that attempt to be a "catch-all" of assessing Gaussian characteristics --- tests from the classic literature of normality testing. Especially tests like Kolmogorov-Smirnov (KS) --- which we found in our evaluation (see Appendix D) to perform best  empirically in this setting among popular tests --- use an extremely broad characteristic; KS compares the empirical cumulative density function (CDF) with the reference CDF, which is "broad" in the sense that a wide range of characteristics --- including norms but also various classes of subtleties --- are encompassed by this characteristic.
>
> An important technical detail is that we use the fact that it is \textbf{equivalent} to test the hypothesis of $\mathbf{x} \sim \mathcal{N}(\mathbf{\mu}, \mathbf{\Sigma}), \mathbf{x} \in \mathbb{R}^D$ for a \textbf{single} observation $\mathbf{x}$ (the latent) as it is to test the hypothesis that a collection of size $D$ of univariate observations $\{\epsilon_d\}_{d=1}^D$ follow $\mathcal{N}(0, 1)$, if $\mathbf{\epsilon} = \mathbf{\Sigma}^{-1}(\mathbf{x} - \mathbf{\mu})$ and $\mathbf{\epsilon} = [\epsilon_1, \epsilon_2, \dots, \epsilon_D]$, which we discuss in Section 3.1 of the paper.
> We will make sure this and its implications are clear in the camera-ready version of the paper, in particular that (1) it is easy to deploy univariate normality tests for testing of the latent as long as it is easy to invert $\mathbf{\Sigma}$ (which is trivial for the diagonal matrices used in most models) and (2) that this equivalence indeed ensures that correlations between latent dimensions are also considered, as asked by Reviewer Y6W4.

---

> ### Author Response · Authors · 2024-11-18
>
> We have addressed individual questions of the reviewers separately. Once again, we thank the reviewers for their feedback.
>
> We believe that the updates we have made to the paper during the rebuttal period based on the feedback from the reviewers --- in particular the updated Figure 4 and 5 --- greatly improves the clarity and the potential impact of our paper.
>
> We look forward to the opportunity to present our improved work.

---

### Comment · Area_Chair_GeXv · 2024-11-24

Dear Reviewers,

This is a gentle reminder that the authors have submitted their rebuttal, and the discussion period will conclude on November 26th AoE. To ensure a constructive and meaningful discussion, we kindly ask that you review the rebuttal as soon as possible and verify if your questions and comments have been adequately addressed.

We greatly appreciate your time, effort, and thoughtful contributions to this process.

Best regards,
AC

---

### Meta-Review · Area_Chair_GeXv · 2024-12-30

**Metareview:**

This paper introduces COG (Combination of Gaussian variables), a method for performing linear combinations of latent variables in generative models while preserving their Gaussian distributional properties. The key claim is that successful generation requires latent vectors to match broad Gaussian characteristics beyond just having likely norms, which they validate through normality testing. The authors show that COG maintains performance comparable to baselines like spherical interpolation for two-point interpolation while enabling new capabilities like defining meaningful latent subspaces - something not previously possible with existing methods. The reviewers appreciated these capabilities and their empirical validation along with theoretical grounding. There were also weaknesses on limited quantitative validations, fair comparisons with baselines (eg, FID computation, spherical interpolation), and human evaluations. Overall, majority of reviewers support acceptance.

**Additional Comments On Reviewer Discussion:**

See above.

---

### Decision · Program_Chairs · 2025-01-22

Accept (Poster)